# Concurrent diffusion of nicotinic acetylcholine receptors and fluorescent cholesterol disclosed by two-colour sub-millisecond MINFLUX-based single-molecule tracking

Francesco Reina [1,2,6,7], Lucas A. Saavedra [3,7], Christian Eggeling [1,2,4,5] ✉ & Francisco J. Barrantes [3] ✉

The diffusion and interaction dynamics of membrane proteins and lipids are key for cell function, but their disclosure is hampered by limited temporal and spatial resolution of conventional observation technologies. Here we exploit the capabilities of minimal fluorescence emission photon fluxes (MINFLUX) microscopy in single-molecule co-tracking experiments of an important membrane protein and cholesterol with enhanced spatiotemporal resolution. Specifically, we interrogate the 2D translational mobility of a ubiquitous cell-surface protein, the nicotinic acetylcholine receptor, in tandem with a fluorescent cholesterol analogue for minute-long periods, reaching nanometric precision and sub-millisecond time resolution. To this end, we implement a multiplexing procedure that enables the simultaneous excitation of the two fluorescent-labelled molecules using a single wavelength, followed by discrimination of their emissions via differential ratiometric recording. We disclose a cholesterol-dependent heterogeneous spectrum of diffusive behaviours with regions of joint translational motion.

Superresolution microscopy methods are increasingly being successfully applied to unravel a wide range of biologically relevant topics[1,2]. Neurobiology has benefited from the inception of these approaches[3], enhancing the spatial and temporal resolution with which we can address the structure and dynamics of the neuron, the patterned compartmentalisation of the axon, and the topography of the macromolecular constituents of the synapse in health and disease[4,5].

Single-particle tracking (SPT) or single-molecule tracking (SMT) techniques are among the biophysical methods of choice to elucidate the spatial extension and temporal duration of dynamic molecular processes involved in protein and cellular function at large. Because of the time window and constrained volume in which they occur, cell-surface phenomena like those involving neurotransmitter receptors are particularly challenging to study. The combination of single-molecule localisation microscopy (SMLM)[6,7], DNA-PAINT

[1]Institute of Applied Optics and Biophysics, Friedrich-Schiller-Universität Jena, Jena, Germany. [2]Leibniz Institute of Photonic Technologies, Jena, Germany. [3]Molecular Neurobiology Division, Biomedical Research Institute, UCA-CONICET, Buenos Aires, Argentina. [4]Jena Centre for Soft Matter (JCSM), Jena, Germany. [5]Leibniz Centre for Photonics in Infection Research (LPI), Jena, Germany. [6]Present address: Max Perutz Labs, Department of Structural and Computational Biology, University of Vienna, Vienna, Austria. [7]These authors contributed equally: Francesco Reina, Lucas A. Saavedra. ✉e-mail: christian.eggeling@uni-jena.de; francisco_barrantes@uca.edu.ar

microscopy[8,9], or the minimal fluorescence emission photon fluxes (MINFLUX) microscopy technique[9–15] with SMT methods is beginning to unravel the dynamics of cellular components.

MINFLUX microscopy, one of the most recent superresolution microscopy methods, combines concepts from stimulated emission microscopy (STED)[16], a targeted approach, and a stochastic localisation superresolution technique like SMLM[17,18]. This powerful combination drastically reduces by a factor of ~100-fold the number of photons needed to localise individual molecules in comparison to camera-based imaging techniques[10,11]. MINFLUX-based SMT is accomplished by employing a patterned doughnut-shaped illumination scheme that initially probes around the estimated emitter position, followed by a computationally driven rapid refocusing of the patterned illumination, zooming in to the fluorescently labelled single molecule to "corral" and "lock" it to follow its trajectory.

Previous studies using camera-based fluorescence microscopy of α-bungarotoxin (BTX)-labelled nAChR have revealed its complex and heterogenous diffusion in the millisecond time window, sensitive to the cholesterol content of the plasma membrane[19–21]. Because of its inherently superior spatiotemporal resolution, MINFLUX microscopy has provided more detailed information on the dynamics of motor proteins[9,22,23] and nuclear pore structure[24] and transport[25]. So far, however, MINFLUX has not been employed for the study of molecular dynamics in live-cell membranes, where the analysis of SMT is challenged by the very fast motion of membrane molecules[26]. Further, and more importantly, until recently, it was only possible to continuously localise one fluorescent emitter at a time with this technique, thus precluding molecule co-tracking studies. As a remedy, the combination of fluorescence energy transfer (FRET) and MINFLUX microscopy as a means to explore short intermolecular distances was suggested by Hell and coworkers[11]. Experimentally, a recent approach for observing two fluorescent emitters at a time combined the use of pulsed MINFLUX with FRET[27]; in a proof-of-concept work, co-tracking of DNA origami labelled with two dyes differing in their fluorescence lifetimes was achieved[28]. Here, we overcome these issues by introducing a simple multiplexing variant in continuous excitation MINFLUX microscopy, using single-wavelength excitation and a ratiometric differential emission detection. In this manner, we exploit the enhanced spatiotemporal resolution and minimal photon budget of MINFLUX to simultaneously follow with nanometric precision and sub-millisecond time resolution the joint single-molecule trajectories of a fluorescent cholesterol analogue and the nAChR protein, affording direct visualisation of interactions between the neurotransmitter receptor and the neutral lipid at the cell surface of a live mammalian cell.

## Results

### Two-colour MINFLUX sub-millisecond single-molecule tracking (SMT) of nicotinic receptor and cholesterol

CHO-K1/A5 cells are a clonal cell line robustly expressing adult muscle-type nicotinic acetylcholine receptor (nAChR)[29]. A set of cells was stained with very low (500 pM) CF®640-labelled α-bungarotoxin (CF®640-BTX). After initial inspection of the cells in the confocal mode, multiple single-molecule tracks were recorded at a maximal rate of ~10 kHz (see "Methods") from at least 10 cells per experimental condition within a usually 10 $\mu m^2$ large region-of-interest (ROI). This allowed us to accumulate in the order of several thousand single-molecule tracks within a few minutes (Fig. 1a). The localisation precision was calculated to be $\sigma = 7 \pm 1$ nm, and the time resolution was $492 \pm 780$ μs (see "Methods"). A second experimental dataset consisted of CHO-K1/A5 cells supplemented with 1 nM of a fluorescent cholesterol analogue, which was conjugated with the organic dye Abberior STAR Red via a polyethene glycol linker (fPEG-Chol). For the co-diffusion experiments, we double-stained the CHO-K1/A5 cells with fluorescent α-bungarotoxin and fPEG-Chol. The spectral overlap of fluorescence emission between the two fluorophores Abberior STAR

Red (labelling fPEG-Chol) and CF®640-BTX (labelling nAChR) was too high to distinguish them by colour. We therefore switched to an α-bungarotoxin labelled with a different fluorophore, CF®680 R (CF®680R-BTX), which emits a more reddish colour, more easily distinguishable from fPEG-Chol. This choice enabled us to excite the two probes with the same laser line at a single wavelength (640 nm) and discriminate between the two by ratiometric measurement of their fluorescence emission wavelengths (CF®680R-BTX, emission maximum, 701 nm; fPEG-Chol, emission maximum: 655). To this end, we established a detector channel ratio (DCR) criterion, as described in Supplementary Fig. 2. We denoted fPEG-Chol trajectories as fPEG-Chol (+CF®680R-BTX) when we analysed the trajectories of the cholesterol probe in the presence of CF®680R-BTX and, reciprocally, CF®680R-BTX (+fPEG-Chol) when we followed CF®680R-BTX trajectories in the presence of fPEG-Chol (Fig. 1b, c). Control experiments with CF®680R-BTX alone or with fPEG-Chol alone were also carried out to determine and denote the DCR values of each individual probe. Our experimental conditions minimise photo-induced biasing effects such as unwanted loss or rise of fluorescence signal, slowdown or speed-up of diffusion and cell death due to photobleaching, photoblueing, photoblinking, phototoxicity, or local heating and trapping, as described in detail in the Supplementary Information.

### Trajectory characterisation and exclusion of immobile trajectories

Preliminary analysis of the data involved measuring trajectory and individual step durations (Supplementary Table 1), which, however, did not reveal further details of the diffusional behaviour of the probes. Since we wanted to disclose the actual (co-) diffusion modes of toxin-labelled nAChR and fluorescent cholesterol, and in view of the biases for completely immobile events in general SMT tracks (including MINFLUX-based)[26], we focus here only on completely mobile trajectories and excluded immobile trajectories from subsequent analyses as in previous work[20,21]. To this end, the criterion of ref. 30 was applied next to the determination of the proportion of immobile/mobile trajectories, as shown in Supplementary Fig. 3. Using this criterion, about half of the trajectories of toxin-labelled nAChRs were classified as completely immobile. Notably, only ~20% completely immobile trajectories were observed in the case of fPEG-Chol tracks co-labelled with CF®680R-BTX.

### Quantitative analysis of nAChR diffusive behaviour

The ensemble-averaged, time-averaged mean squared displacement (EA-TA-MSD) is a time-dependent measure of the deviation of a particle position relative to a reference position. Here, EA-TA-MSD curves (Fig. 1d) were furnished by fitting the individual curves with the analytical expressions (Eqs. 3–4) outlined in "Methods", providing information on the ensemble diffusive behaviour of the nAChR macromolecules and the cholesterol analogue in a 2D Euclidean space —the membrane—over time.

MINFLUX allowed us to analyse a hitherto unexplored time window, i.e., the sub-millisecond time range (see Fig. 1d). As a comparison, previous camera-based studies had a temporal resolution of ≈ 10 ms[20,21]. Table 1 and Supplementary Table 2 list the kinetic parameters derived from this analysis, i.e., the generalised diffusion coefficient ($K_\beta$) and the apparent anomalous coefficient (β). One would expect no significant differences between the translational diffusion properties of CF®640R-BTX and CF®680R-BTX-labelled nAChRs, but when averaged over the *entire ensemble population* of tracks, the nAChRs labelled with the two different toxins appeared to differ in their kinetics (Table 1). However, as shown below (Table 2 and Supplementary Material), due to a few outliers, this difference was only apparent and not statistically significant.

Table 1 also lists the average duration (or residence time) of confinement sojourns and non-confined portions, respectively, and

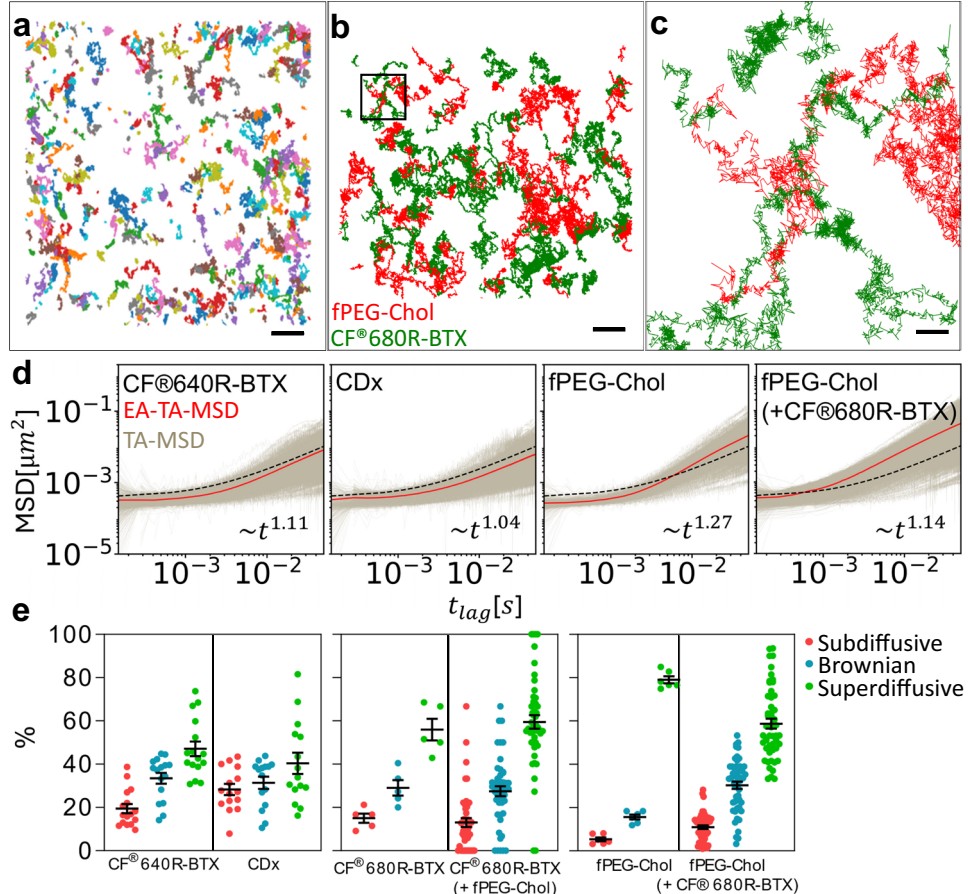

**Fig. 1 | MINFLUX single-molecule tracking of individual bungarotoxin-labelled nicotinic acetylcholine receptors and fluorescent cholesterol in live cells, ensembled-averaged (EA), time-averaged (TA) mean squared displacement (EA-TA-MSD) and TA-MSD analyses, 2D-diffusion parameters and diffusive regimes derived from the anomalous coefficient β. a** Example of a 10 ×10 μm region of interest (ROI) image of the clonal cell line CHO-K1/A5 expressing adult muscle-type nAChRs labelled with CF®640R-BTX only. Single-molecule tracks are randomly colour-coded as they appear on the screen of the MINFLUX microscope setup; after a few minutes, the ROI is covered with individual traces. Approximately 50% of the fluorescently labelled receptors appeared as nanometric-sized dots with no indication of motion. Subsequent analysis identified them as immobile nAChRs (Supplementary Fig. 3). Scale bar = 1 μm. **b** Representative fPEG-Chol and CF®680R-BTX-labelled nAChR trajectories in a co-tracking experiment. Scale bar = 500 nm.

**c** Zoom-in into marked region from (**b**). Scale bar = 100 nm. **d** MSD analyses from different samples as marked (CDx: cholesterol depletion via methyl-β-cyclodextrin), TA-MSDs, EA-TA-MSD, and ideal Brownian diffusion (see Eq. 5 in "Methods", where $R$, the correction factor for blurring, was set to $1/6.2^{10}$, and $\Delta = 10$ nm). Both *x*- and *y*-axes are in log scale. **e** Boxplot showing the percentage of trajectories classified as subdiffusive (β < 0.9), Brownian (0.9 ≤ β ≤ 1.1), and superdiffusive (β > 1.1) for CF®640R-BTX ($n = 1383$ trajectories with an average of 1105 steps, 16 ROIs), cholesterol-depleted CF®640R-BTX-labelled samples (CDx, $n = 853$ and average 793 steps, 15 ROIs), CF®680R-BTX ($n = 187$ and average of 925 steps, 5 ROIs), CF®680R-BTX (+fPEG-Chol) ($n = 1373$ and average of 2034 steps, 47 ROIs), fPEG-Chol ($n = 1012$ and average of 1363 steps, 6 ROIs), and fPEG-Chol (+CF®680R-BTX) ($n = 5229$ and average of 1496 steps, 52 ROIs). TA-MSD and EA-TA-MSD fittings extended up to 50 ms. Error bars represent standard errors of mean (SEM).

the confined ratio $r$, which is defined as the quotient between the residence time of the trajectory in the confined state (i.e., the sum of the confined portions of a single trajectory) and the total duration of the trajectory for the ensemble population of trajectories. The most striking observation is the modification of the kinetic parameters of each probe in the presence of the other (Table 1). Thus, fPEG-Chol diffused ~1.4-fold faster in the presence of the toxin-labelled nAChR (fPEG-Chol (+CF®680R-BTX) samples; $p < 0.01$), as did the toxin in the presence of the cholesterol analogue ($p < 0.01$).

The power (anomalous) exponent β obtained by fitting the individual time-averaged MSD (TA-MSD) curves was next used to classify trajectories according to their motional regime, as shown in Fig. 1 and Table 2. Tracks with β < 0.9 were rated as subdiffusive, i.e., mobility was transiently confined or trapped, with 0.9 ≤ β ≤ 1.1 as Brownian, i.e., free diffusion, and β > 1.1 as superdiffusive, i.e., deviating from free diffusion (due to enhanced mobility owing to, e.g., active cellular transport processes). Both the fluorescent-labelled nAChR and the cholesterol analogue covered a wide spectrum of diffusive behaviours,

as shown in Fig. 1e. CF®640R-BTX- or CF®680R-BTX-labelled nAChRs trajectories exhibited a predominantly (~47–60%) superdiffusive component, ~33% Brownian, and ~20% subdiffusive ($p < 0.05$). fPEG-Chol depicted a completely different, overwhelmingly (~80%; $p < 0.0001$) superdiffusive behaviour, which diminished in samples co-labelled with CF®680R-BTX (~60 %; $p < 0.05$), concomitant with an increase in subdiffusive ($p < 0.05$) and Brownian ($p < 0.001$) trajectories. The large fraction of superdiffusive trajectories resulted from the threshold set for β (> 1.1) for this population of trajectories, to facilitate comparison with previous SMT analyses[20,21,31]. This enabled us to highlight tendencies towards faster diffusion, e.g., due to larger fractions of more fluid (or less ordered) membrane regions during the track. When analysed according to their diffusive behaviour, subdiffusive and Brownian particles labelled with either toxin displayed statistically indistinguishable values, while CF®680R-BTX super-diffusive particles moved faster than those of CF®640-BTX. This is clearly observed in Supplementary Fig. 8, where the histograms showing the distribution of the $K_\beta$ values of the two fluorescent toxins

**Table 1 | Average generalised diffusion coefficient $K_\beta$, apparent anomalous coefficient (β) of far-red fluorescent α-bungarotoxin (CF®680R-BTX) and fluorescent cholesterol (fPEG-Chol) averaged over all trajectories, average durations of confined and non-confined portions, and their ratio (r)**

| Experimental condition | Mean ± S.E.M. | | | | |
| --- | --- | --- | --- | --- | --- |
| | $K_\beta$ $(\mu m^2 s^{-\beta})$ | β | Average duration of confinement sojourn [ms] | Average duration of non-confined portion [ms] | [a]Confined ratio, r |
| Far-red toxin (CF®680R-BTX)-labelled nAChR alone | 0.428 ± 0.017 | 1.16 ± 0.02 | 66 ± 8 | 20 ± 2 | 0.63 ± 0.01 |
| Cholesterol analogue (fPEG-Chol) alone | 1.357 ± 0.082 | 1.26 ± 0.01 | 25 ± 2 | 9 ± 1 | 0.63 ± 0.01 |
| Far-red toxin (CF®680R-BTX)-labelled nAChR in the presence of fPEG-Chol | 0.926 ± 0.052 | 1.14 ± 0.02 | 39 ± 5 | 10 ± 1 | 0.67 ± 0.01 |
| fPEG-Chol in the presence of far-red (CF®680R-BTX) toxin | 1.710 ± 0.045 | 1.13 ± 0.01 | 8 ± 2 | 7 ± 1 | 0.49 ± 0.01 |

[a]Confined ratio r is defined as the quotient between the residence time of the trajectory in the confined state divided by the total duration of the trajectory.

**Table 2 | Dissection of the anomalous exponent β and generalised diffusion coefficient $K_\beta$ for each of the three diffusion regimes (subdiffusive, Brownian and superdiffusive) and their percentage**

| Anomalous exponent β and generalised diffusion coefficient $K_\beta$ | | | | | | | | | |
| --- | --- | --- | --- | --- | --- | --- | --- | --- | --- |
| Experimental condition | Subdiffusive | | | Brownian | | | Superdiffusive | | |
| | % | $K_\beta$ | β | % | $K_\beta$ | β | % | $K_\beta$ | β |
| CF®640-BTX | 19.44 ± 2.14 | 0.062 ± 0.002 | 0.78 ± 0.01 | 33.47 ± 2.50 | 0.149 ± 0.005 | 1.00 ± 0.01 | 47.09 ± 3.37 | 0.530 ± 0.02 | 1.28 ± 0.01 |
| CF®640-BTX upon cholesterol depletion (CDx) | 28.28 ± 2.56 | 0.051 ± 0.002 | 0.77 ± 0.01 | 31.34 ± 2.83 | 0.118 ± 0.006 | 0.97 ± 0.01 | 40.39 ± 4.91 | 0.379 ± 0.031 | 1.31 ± 0.01 |
| CF®680-BTX | 15.01 ± 2.14 | 0.062 ± 0.006 | 0.80 ± 0.01 | 29.06 ± 3.58 | 0.130 ± 0.010 | 1.01 ± 0.01 | 55.92 ± 5.00 | 0.662 ± 0.074 | 1.34 ± 0.02 |
| fPEG-Chol | 5.34 ± 0.89 | 0.124 ± 0.007 | 0.77 ± 0.01 | 15.65 ± 1.12 | 0.292 ± 0.011 | 1.02 ± 0.01 | 79.01 ± 1.62 | 1.689 ± 0.057 | 1.33 ± 0.01 |
| CF®680-BTX (+fPEG-Chol) | 13.01 ± 2.01 | 0.095 ± 0.006 | 0.74 ± 0.01 | 27.50 ± 2.31 | 0.345 ± 0.014 | 1.01 ± 0.01 | 59.50 ± 3.12 | 1.475 ± 0.069 | 1.29 ± 0.01 |
| fPEG-Chol (+CF®680-BTX) | 10.96 ± 0.86 | 0.353 ± 0.005 | 0.77 ± 0.01 | 30.30 ± 1.66 | 0.868 ± 0.009 | 1.01 ± 0.01 | 58.74 ± 2.25 | 2.482 ± 0.033 | 1.26 ± 0.01 |

are displayed: only a minor proportion (~1%) of outliers corresponding to the superdiffusive tracks of CF®680R-BTX-labelled receptors are singled out. Moreover, the median of the two distributions was 0.18 and 0.19 for CF®640R-BTX and CF®680R-BTX, respectively, also not statistically different. It is therefore the weight of the superdiffusive component that distorted the *apparent average value of the ensemble population*. Moreover, regardless of whether CF®640R-BTX or CF®680R-BTX (+fPEG-Chol) was used, both probes exhibited accelerated diffusion with increasing concentrations of exogenous cholesterol, and the trend remained consistent for the other dynamic parameters (Table 1 and Supplementary Table 2).

**Dissection of trajectory properties in non-confined (free walks) and confined states**

The dynamics within the free and confined states of the trajectories were next dissected separately (Fig. 2a). The free, non-confined segments of CF®640R-BTX-labelled nAChR lasted 12 ± 1 ms, while the confined segments lasted significantly longer (71 ± 8 ms) (Fig. 2b). The presence of fPEG-Chol in co-labelled samples significantly decreased the duration of both non-confined (20 ± 1 ms to 10 ± 1 ms, $p < 0.05$) and confined (66 ± 8 ms to 39 ± 5 ms, $p < 0.05$) states of CF®680R-BTX-labelled nAChRs. Cholesterol depletion (CDx) had the opposite effect, lengthening the duration of the confined portions to 137 ± 25 ms ($p < 0.01$). This was confirmed by experiments in which unlabelled cholesterol was added to the toxin-labelled samples, where the duration of the confined portions was reduced to a similar extent as that produced by fPEG-Chol (27 ± 6 ms at 50 nM, $p < 0.01$; Supplementary Fig. 4).

To quantitatively assess the time spent by a trajectory in the confined state, a "confined ratio", r (Table 1 and Fig. 2b), was operationally defined as the quotient between the residence time of the

trajectory in the confined state divided by the total duration of the trajectory. r was 0.76 ± 0.02 and 0.63 ± 0.03 for CF®640R-BTX and CF®680R-BTX, respectively, indicating that, on average, nAChRs spend most of their lifetime in confinement, as was the case with fPEG-Chol ($r = 0.63 ± 0.02$) alone. The latter value decreased to 0.49 ± 0.02 in cells co-labelled with toxin (fPEG-Chol (+CF®680R-BTX), $p < 0.01$).

Intuitively, if a molecule spends less time under confinement, one expects it to spend more time in the complementary non-confined part of the track. However, both non-confined and confined portions of CF®680R-BTX trajectories were shortened under fPEG-Chol co-labelling conditions (Fig. 2). A possible explanation for this paradoxical observation is that the *transition rate* (i.e., the number of confined/non-confined state changes per unit time) (Fig. 2b, right) increased in the presence of cholesterol. Indeed, the transition rate of CF®680R-BTX increased from 25.49 ± 1.93 $s^{-1}$ to 48.27 ± 2.99 $s^{-1}$ in the presence of fPEG-Chol ($p < 0.01$). The transition rate of fPEG-Chol alone was 62.94 ± 4.79 $s^{-1}$, augmenting to 103.70 ± 2.68 $s^{-1}$ in the presence of CF®680R-BTX ($p < 0.001$). These results indicate that CF®680R-BTX changed state faster in the presence of fPEG-Chol and vice versa. Finally, supplementation with unlabelled cholesterol (50–100 nM) did not change the *total* time that the nAChR spent exploring each state, but rather changed the average residence time in confinement, which decreased from 0.84 ± 0.09 s to 0.34 ± 0.09 s upon cholesterol supplementation.

We next analysed the free-moving and confined portions of the trajectories more closely by computing the TA-MSD of each portion[32], i.e., we determined values of the generalised diffusion coefficient ($K_\beta$) and the apparent anomalous coefficient (β) for the non-confined and confined portions of the trajectories. The values are listed in Supplementary Table 3. The values of β again allowed us to rate the trajectory portions as either subdiffusive (β < 0.9), Brownian (0.9 ≤ β ≤ 1) or

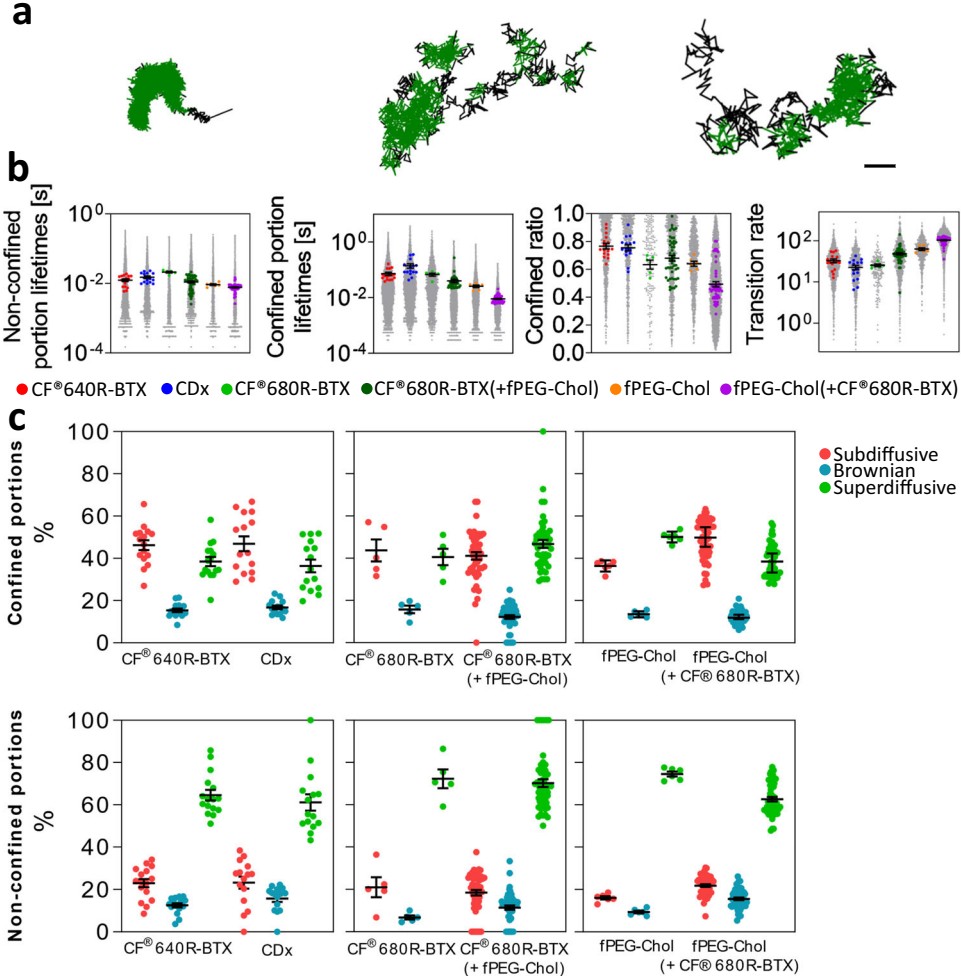

**Fig. 2 | Confined and non-confined states of the cholesterol probe and toxin-labelled nAChR trajectories and their corresponding diffusive behaviour.**
**a** Representative single tracks of CF®640R-BTX-labelled nAChR under control conditions showing parts classified as confinement sojourns (green) and as free, non-confined portions (black). Scale bar = 100 nm. **b** Boxplots showing the mean and SEM of lifetimes of non-confined portions (left) and confinement sojourns (second left), confined ratio $r$ (second right, defined as before $r$ = confined portion lifetimes / total trajectory time), and transition rates between confined and non-confined states of the trajectories of CF®640R-BTX (19,427 confined portions, 19,443 non-confined portions, and 1383 trajectories in 16 ROIs), CDx (9169 confined portions, 9099 non-confined portions, and 853 trajectories in 15 ROIs), CF®680R-BTX (2458 confined portions, 33,064 non-confined portions, and 187 trajectories in 5 ROIs), CF®680R-BTX (+fPEG-Chol) (32,881 confined portions, 21,923 non-confined portions, and 1373 trajectories in 47 ROIs), fPEG-Chol (18,210 confined portions, 18,300 non-confined portions, and 1012 trajectories in 6 ROIs), and fPEG-Chol (+CF®680R-BTX) (161,601 confined portions, 21,923 non-confined portions, and 5229 trajectories in 52 ROIs). Each grey dot represents a trajectory inside an ROI, and the coloured dots represent the average of the parameter for a given ROI. **c** Boxplots showing the mean and SEM of the percentage of trajectories in each experimental ROI classified as subdiffusive ($\beta < 0.9$), Brownian ($0.9 \leq \beta \leq 1.1$), and superdiffusive ($\beta > 1.1$) in confinement sojourns (upper panels) and non-confined portions (lower panels), respectively, upon TA-MSD fitting up to 25 ms for the different sample conditions as labelled. Source data are provided as a Source Data file.

superdiffusive ($\beta > 1.1$), and determine the respective percentages, as done before for the whole tracks (compare Figs. 1e and 2c). As shown in Fig. 2c, the percentage of subdiffusive motion was higher in the confinement sojourns, and, in the case of fPEG-Chol and CF®680R-BTX, the predominant behaviour in the non-confined state was in all cases superdiffusive. Both behaviours were to be expected, since confinement is a form of subdiffusion, while non-confinement is characterised by fast diffusion. The values of $K_\beta$ followed similar trends: in the non-confined state of the nAChR, $K_\beta$ was ~3 times faster in the presence of fPEG-Chol ($p < 0.05$). Cholesterol depletion or supplementation modified the diffusion properties in both confined and non-confined regions, as highlighted in Supplementary Table 3 and Supplementary Fig. 4. Finally, Supplementary Fig. 5 depicts additional metrics of the trajectories, such as the shape and size of a confinement area (in terms of the total area, eccentricity and length of the major axis), and the number of observed steps of the trajectories within and between

confinement sojourns. Further details on all the parameters are discussed next to the respective Supplementary Figs. and tables.

## Turning angle analysis
The high spatial and temporal resolution of the tracks, as afforded by MINFLUX microscopy, allowed us to use another metric to characterise the diffusional behaviour of nAChR and fluorescent cholesterol in their confined and non-confined states: the turning angles distended by the molecules along their SMTs. Specifically, we calculated the probability density function of turning angles between subsequent steps of a trajectory for different step lengths, as shown in Fig. 3 for the unclassified and in Supplementary Fig. 6 for the differently classified diffusion regimes (subdiffusive, Brownian, superdiffusive). The probability density function for Brownian motion is constant for angles between 90° to 180° (since there is no turning angle predilection in this case), while there is an increase towards turning angles between 90°

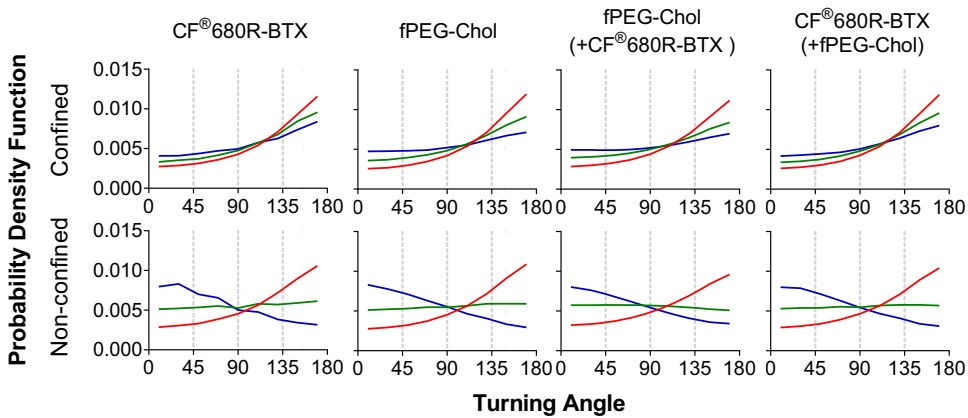

**Fig. 3 | Breakdown of turning angle analysis of nAChR and cholesterol analogue trajectories discriminated into confined and non-confined, free-walking states.** Probability function (PDF) of the turning angles (0° to 180°) of the trajectories for the different samples and the two classifications as labelled. The differently coloured probability density functions correspond to 1 (*red*), 4 (*green*), 8 (*blue*) step lags. Source data are provided as a Source Data file.

and 180° (i.e., anticorrelation) for confined or subdiffusive characteristic (due to molecules tending to move backwards), and turning angles between 0° and 90° (i.e., a positively correlated steps) for superdiffusive behaviours (as the molecule tends to move forward). Further, changes in the probability density function distribution for increased step lengths indicate the spatial extent of a confined, free-diffusive or superdiffusive portion of a track.

The turning angle analysis confirmed our previous analysis. The distribution for both CF®680R-BTX-labelled nAChRs and fPEG-Chol within confinement sojourns increased linearly as the turning angle approached 180°, i.e., both molecules tended to undergo anticorrelated steps. This confirms their prominently subdiffusive and rare Brownian-like or superdiffusive behaviour, as already depicted in Fig. 2. Moreover, this preference for anticorrelated steps diminished as the step lag increased, suggesting the finite extent of confinement zones, thus hindering diffusion inside structures of characteristic size[21,33]. This finding was further supported by deep learning CONDOR analysis[34], as shown in Supplementary Fig. 7 and Supplementary Table 5; restricted mobility of both toxin-labelled nAChR *and* fPEG-Chol occurred predominantly within membrane compartments of specific sizes (0.006–0.008 μm²).

In non-confined portions of the tracks, for short step lags, the motion was typical of obstructed diffusion[35]; for intermediate length steps, the turning angle distribution was uniform, characteristic of Brownian motion[33]; and for long step lags, the molecules took highly correlated, i.e., superdiffusive steps. This change in directionality of the turning angle distributions with variation of the step lag confirmed, on one hand, the heterogeneous characteristic of the diffusion even in the non-confined portions of the SMTs, and, on the other hand, the augmented superdiffusive motion in the non-confined state, as shown in Fig. 3.

**Temporal overlap between nAChR and cholesterol trajectories**
In a next step, we analysed simultaneously recorded trajectories of nAChR (labelled with CF®680R-BTX) and fPEG-Chol (labelled with Abberior STAR Red) probes with high detail, especially to disclose any co-diffusion events. As highlighted before, this choice of labels (CF®680R and Abberior STAR Red) enabled us to excite the two probes with the same laser line and discriminate between the two by ratiometric measurement of their fluorescence intensities in two spectrally separated detection channels. Specifically, we calculated the detector channel ratio (DCR) by division of the fluorescence intensities registered in the more reddish and blueish detection channels, respectively (see "Methods" and Supplementary Fig. 2). Control measurements on

the two probes individually revealed clearly separated high and low DCR values for fPEG-Chol and CF®680R-BTX, respectively (Supplementary Fig. 2). This is further depicted in the left two panels of Fig. 4a, which highlight representative tracks and the temporal evolution of DCR and total intensity values summed over both detection channels and given as the effective counts at offset (ECO). The latter correspond to the photon counts collected at the outer points of the MINFLUX scanning pattern, corrected for background contributions for individually recorded fPEG-Chol and CF®680R-BTX. DCR values were above 0.55 in the case of CF®680R-BTX and below 0.40 for fPEG-Chol, and ECO/intensity values were at very low levels. However, simultaneous recordings of both probes revealed a significant fraction of trajectories with DCR values between 0.4 and 0.55 and at the same time increased ECO/intensity values, disclosing sections of simultaneous co-tracking detection and thus co-diffusion of the two probes (Fig. 4a). This is also revealed in the frequency histograms of the DCR values, revealing a shift of the DCR values towards the intermediate regime between 0.4 and 0.55 (Fig. 4b). Figure 4b also clearly highlights that both modulation of the actin cytoskeleton by CK-666 and addition of the unlabelled cholesterol affected the co-tracking of the nAChR and fluorescent cholesterol.

**Quantitative assessment of spatial overlap between toxin-labelled nAChR and fPEG-Chol trajectories**
Through visual inspection, we observed that trajectories of fPEG-Chol spatially intersected with those of CF®680R-BTX in confinement and non-confined zones, in several cases more than once along the same trajectory (Fig. 5a, left). In fact, we found that in a majority (~70%) of experimental ROIs, there were statistically significant overlaps in the confined portions of the trajectories of nAChR and cholesterol (*p* < 0.01). This effect was quantified by the overlap coefficient metric (*C*) herewith introduced (see "Methods"). For the trajectories analysed in this study, we calculated a value *C* = 0.27 ± 0.02, or, in other words, 27% of the confinement areas detected in CF®680R-BTX trajectories, which target the nAChR, overlap with those of fPEG-Chol. Addition of 100 nM unlabelled cholesterol drastically reduced the overlap by more than two-thirds (*C* = 0.08 ± 0.01, *p* < 0.001). The overlap of non-confined portions was also found to be statistically different from chance: 5% of the ROIs were found to overlap (*p* < 0.01) along the free-walk portions, with a *C* value of 0.40 ± 0.03, indicating that the degree of overlap in non-confined portions was higher than in confined portions. Moreover, upon addition of 100 nM cholesterol, the non-confined overlap coefficient C was more drastically reduced to 0.13 ± 0.01 (*p* < 0.001), suggesting that the unlabelled cholesterol

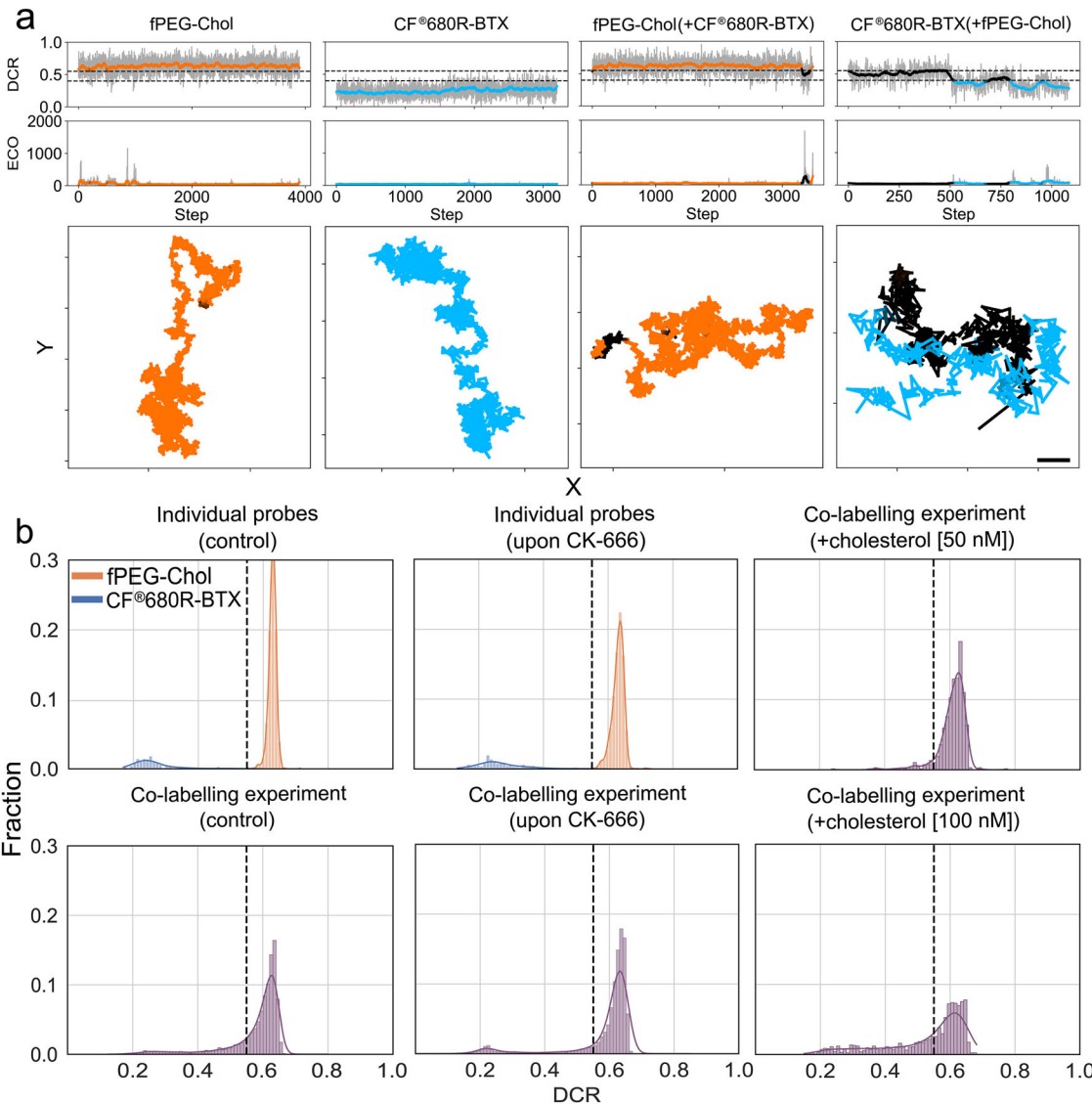

**Fig. 4 | Co-diffusion analysis of fPEG-Chol and CF®680R-BTX-labelled nAChR using detector channel ratio (DCR) discrimination in combination with intensity changes. a** Representative data of DCR and intensity values (as effective counts at offset (ECO)) over time steps (upper two panels) and corresponding tracks in the X-Y plane with colours indicating classification as fPEG-Chol (orange), CF®680R-BTX (blue) and co-diffusion (black) for cells with fPEG-Chol alone, CF®680R-BTX nAChR alone, and the 2-colour co-labelling experiments with both fPEG-Chol and CF®680R-BTX. Co-diffusion becomes apparent in the increased ECO values and simultaneous change in DCR values converging in a corridor between 0.40 and 0.55 (marked by the two horizontal dashed lines). Scale bar = 100 nm.

**b** Histograms of DCR values for the tracking data of fPEG-Chol (orange) and CF®680R-BTX (blue) only, and of fPEG-Chol and CF®680R-BTX co-labelled cells (purple) with two-species Gaussian fits for experiments with only one probe without (upper left, control) and with (upper middle) actin modulation by CK-666, and with doubly labelled experiments without any treatment (lower left, control), with actin modulation by CK-666 (lower middle) and with cholesterol supplementation (50 nM: upper right, 100 nM: lower right). The histograms built from the data acquired with both probes simultaneously show only the mobile fraction. Source data are provided as a Source Data file.

displaced the fluorescent-labelled fPEG-Chol, resulting in less detectable co-diffusion events.

## Perturbation of the actin subcortical meshwork

To investigate the influence of the (cortical) actin cytoskeleton on the (co-)diffusion dynamics of nAChR and cholesterol, we incubated the samples of CHO-K1/A5 cells for 15 min at 4 °C with CK-666, a compound that inhibits Arp2/3 complex formation[36,37] and hence the branching and growth of new actin filaments, thus perturbing the stability of the submembrane (cortical) actin meshwork. The cortical actin cytoskeleton, i.e., the actin filaments underlying the plasma membrane, usually builds up a checkered-like network, "fencing in" compartments, where diffusion over an actin border from one

compartment to the next is restricted, resulting in a compartmentalised or hop-like diffusion (see, e.g., ref. 38). Experiments from our laboratory have also reported effects of actin cytoskeleton perturbation on nAChR organisation[39]. The effects of CK-666 treatment on the kinetic diffusion parameters of CF®680R-BTX nAChR and fPEG-Chol are included in Supplementary Table 2 to facilitate comparison with the other conditions. No changes were apparent in the anomalous diffusion coefficient β for any of the experimental conditions tested, except for fPEG-Chol: upon CK-666 treatment, the fluorescent cholesterol analogue was less superdiffusive ($p < 0.05$). Only subtle changes were observed in the apparent diffusion constant $K_\beta$; the fluorescent cholesterol probe alone was 1.1 times faster, and in the presence of CF®680R-BTX, 1.3 times faster, upon CK-666 treatment.

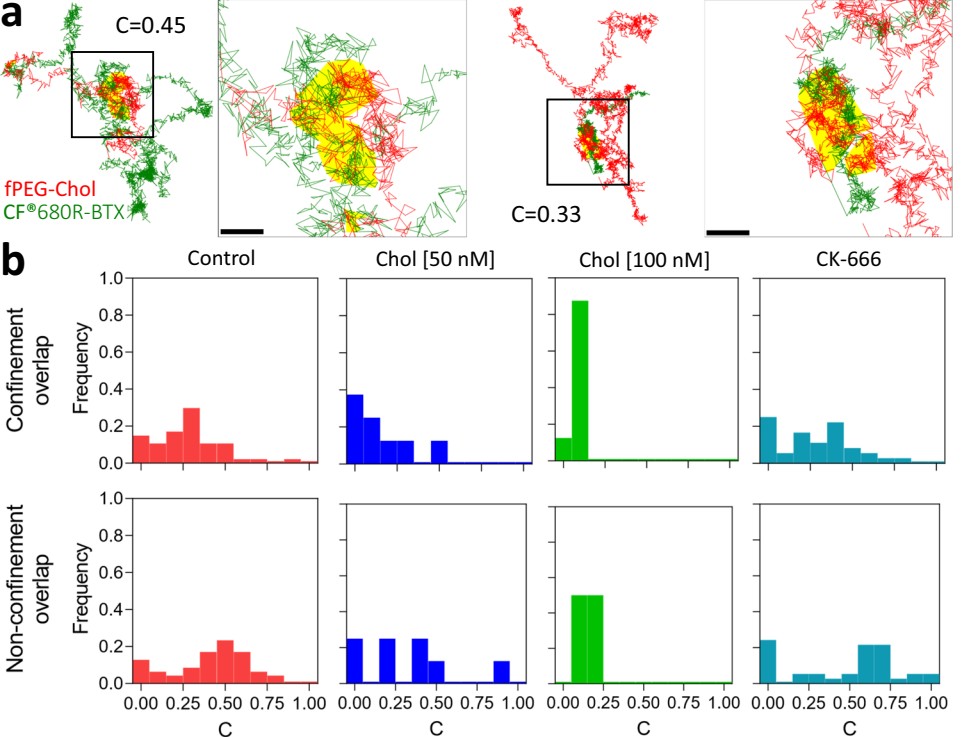

**Fig. 5 | Spatial overlap of BTX-labelled nAChR and fluorescent cholesterol tracks in confinement sojourns and Brownian sections of the tracks, and effects of cholesterol addition and CK-666 treatment. a** Two representative examples of trajectories of the two-colour labelled experiment with the track of the CF®680R-BTX-labelled nAChR and of Abberior STAR Red labelled fPEG-Chol: large overviews (left and middle right panels) and respective zoom-ins to the indicated black boxes (middle left and right). The confinement areas classified for CF®680R-BTX and of fPEG-Chol are highlighted in yellow. Their overlap coefficient C is indicated as a number in each case. Scale bar = 50 nm. **b** Histograms of overlap coefficient C (bin width = 0.1) for CF®680R-BTX and fPEG-Chol without treatment (left), following 50 nM (middle left) and 100 nM (middle right) cholesterol addition, and upon CK-666 treatment (right) in confinement (upper panels) and non-confinement (lower panels) areas.

The confined ratio *r* was not affected by CK-666 treatment, except for CF®680R-BTX in the presence of fPEG-Chol, in which case *r* decreased moderately ($p < 0.01$). The durations of the confined segments of the tracks were also affected: fPEG-Chol trajectories increased, and those of CF®680R-BTX decreased ($p < 0.01$) upon CK-666 treatment (Supplementary Table 2). Interestingly, the drug did not affect the degree of overlap of fPEG-Chol and toxin-labelled nAChR in the confinement sojourns ($C = 0.27 \pm 0.01$) but did increase the degree of overlap in non-confined portions ($0.46 \pm 0.05$; $p < 0.01$).

In order to investigate whether Arp2/3-mediated perturbation of the submembrane actin meshwork induced the so-called hop diffusion, we next undertook an analysis of the trajectories following the approaches of Hell and coworkers[40] and Eggeling and coworkers[41]. Here we classified the motion of molecules in three models: freely diffusing, confined, and hopping between compartments (Fig. 6). For the hopping diffusion we introduced specific kinetic diffusion parameters of the average actin compartment diameter L, the diffusion coefficients $D_\mu$ and $D_M$ characterising the short-term diffusion within the actin compartments (intra-compartmental), and long-term diffusion across compartments (inter-compartmental), respectively, and the average scale length $L_{hop}$ of spatial separation between $D_\mu$ and $D_M$ (Table 3). For both CF®680R-BTX and fPEG-Chol, only a small portion of the trajectories revealed hop-like diffusion (-15%); diffusion was rather dominated by free (39% and 30% for CF®680R-BTX and fPEG-Chol, respectively) and confined (39% and 30% for CF®680R-BTX and fPEG-Chol, respectively) diffusion. CK-666 treatment led to hardly any change in these figures since the actin network was still intact but with less tight branching, i.e., larger compartment sizes, highlighted by the increase in values of *L* and $L_{hop}$ from around 130–160 nm to

170–250 nm. Most significantly, CK-666 treatment resulted in overall increases in the diffusion coefficients: (1) the diffusion coefficient $D_{free}$ of the freely diffusing fraction increased by a factor of 2.8 for CF®680R-BTX and 1.6 for and fPEG-Chol; (2) mobility was also increased for the confined trajectories (a factor of 2.45 for CF®680R-BTX and 1.3 for fPEG-Chol); and (3) for hopping diffusion, both the intra- and inter-compartment coefficients $D_\mu$ and $D_M$ increased by a factor of 2–2.5. Despite this increase in $D_\mu$ and $D_M$, the compartmentalisation strength $S = D_\mu/D_M$, which is inversely proportional to the hopping probability (i.e., the probability of crossing an actin compartment boundary)[41], remained almost constant (3.07–3.46 for CF®680R-BTX and 2.43–3.45 for fPEG-Chol). This indicates that although the velocity of the molecules increased, they remained spatially constrained upon disturbing the cytoskeletal fences with CK-666, as highlighted above when discussing the negligible changes in fractions of diffusion modes and values of *L* and $L_{hop}$.

## Discussion

We have applied MINFLUX nanoscopy to simultaneously follow the lateral motion of individual fluorescent-tagged cholesterol and neurotransmitter receptor protein molecules at a sampling rate of up to 10 kHz, i.e., at ≈600 μs per individual step, and with a localisation precision $\sigma = 7 \pm 1$ nm (Supplementary Table 6). This accomplishment required the development of a novel detection scheme that enabled the co-tracking of both the Abberior STAR Red labelled cholesterol probe (fPEG-Chol) and the nicotinic acetylcholine receptor (nAChR labelled with fluorescent BTX). The fluorescence emission of the two probes was thereby excited simultaneously at the same wavelength and ratiometrically discriminated by emission wavelength,

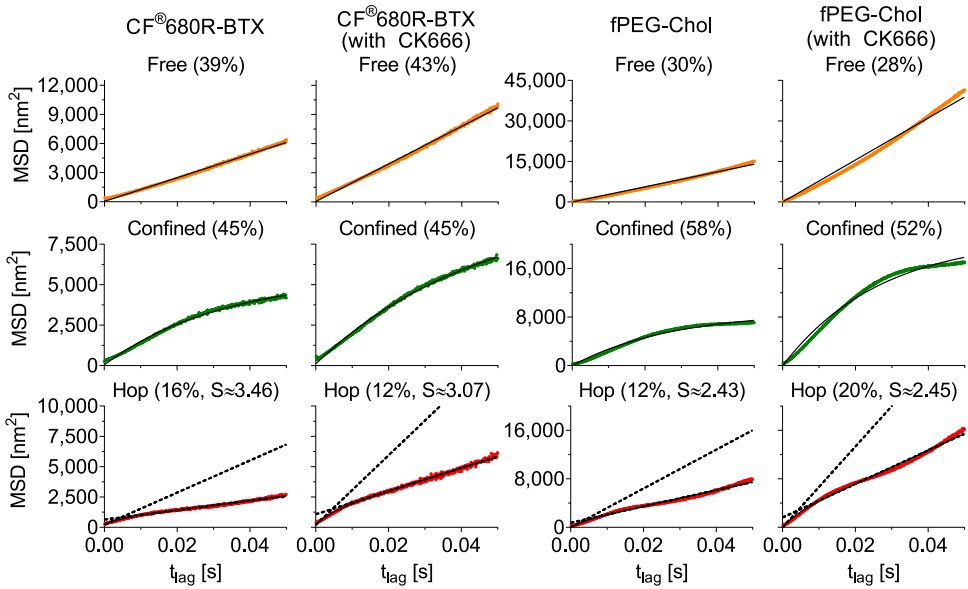

**Fig. 6 | Analysis of diffusion data upon actin modulation.** Ensemble-averaged, time-averaged mean square displacement (EA-TA-MSD) curves determined for all trajectories (Brownian and subdiffusive) of CF®680R-BTX and fPEG-Chol with and without perturbation of the subcortical actin meshwork by CK-666, as labelled. The motional states were classified into three categories: free (orange, upper panels), confined (green, middle panels), and hop-like (red, lower panels) diffusion according to ref. 41 and ref. 40. $S$ denotes the confinement strength for the hopping diffusion, defined as the $D_\mu/D_M$ ratio, and the black lines in the panels of the upper two rows give the fits. The dotted lines in the panels of the lower row correspond to the expected MSD for pure diffusion with $D_\mu$ and $D_M$. Source data are provided as a Source Data file.

**Table 3 | SMT diffusion parameters of toxin-labelled nAChR and fluorescent cholesterol trajectories corresponding to different diffusion models upon actin cytoskeleton perturbation with CK-666**

| Condition | $D_{free}$ [$nm^2s^{-1}$] | $D_\mu$ [$nm^2s^{-1}$] in confinement | $D_M$ [$nm^2s^{-1}$] between compartments | $D_\mu$ [$nm^2s^{-1}$] within compartments | $L$ [$nm$] | $L_{hop}$ [$nm$] |
|---|---|---|---|---|---|---|
| fPEG-Chol | 69,980 | 81,339 | 33,612 | 81,748 | 163.20 | 62.46 |
| fPEG-Chol upon CK-666 treatment | 194,595 | 196,997 | 69,116 | 169,990 | 251.80 | 90.93 |
| Far-red toxin (CF®680R-BTX) | 30,393 | 42,181 | 9770 | 33,842 | 128.26 | 42.05 |
| Far-red toxin (CF®680R-BTX) upon CK-666 treatment | 48,267 | 54,158 | 23,813 | 73,193 | 172.88 | 62.34 |

$D_{free}$: average diffusion coefficient in the trajectories classified as free diffusing. $D_\mu$ in confinement: diffusion coefficient within confined sojourns of the trajectories. $D_\mu$: in between compartments, i.e., short-time (intra-compartmental) diffusion coefficient for the trajectories classified as hopping diffusion. $D_M$: long-time inter-compartmental diffusion coefficient for the trajectories classified as hopping diffusion. $L$: apparent confinement domain size or, in this case, the average actin compartment size explored in the trajectories classified as hopping diffusion. $L_{hop}$: average scale length of separation between $D_\mu$ and $D_M$ in the trajectories classified as hopping diffusion.

overcoming a bottleneck of conventional MINFLUX studies. Single-molecule trajectories of a fluorescent phospholipid, DPPE-ATTO 647 N, in an artificial lipid bilayer[42] and the regular stepping motion of the kinesin protein motor on microtubules have been documented with MINFLUX[9,22,23] or MINSTED[43] but to our knowledge sub-millisecond co-tracking localisation studies of a membrane protein macromolecule together with a fluorescent lipid in the complex membrane environment of a living cell have not been reported to date.

### Diffusion properties of the BTX-labelled nAChRs and the fluorescent cholesterol probe
Exploiting the superior time-resolution capability of the MINFLUX technique, we were able to dissect the diffusional properties of individual trajectories into their non-confined and confined portions along the tracks. nAChRs labelled with either CF®640-BTX or CF®680-BTX displayed a heterogeneous diffusion, ranging from subdiffusive (i.e., confined) over unhindered (i.e., random Brownian) to comparably very fast (i.e., superdiffusive) motion, as reported before in camera-based studies[20]. Subdiffusive motion became more pronounced upon cholesterol depletion, yet with a high proportion of superdiffusive components (Fig. 1 and Table 2). This provided the first piece of evidence on the influence of the neutral lipid on the nAChR motility in this hitherto unexplored temporal window. The simplest explanation for the translational motion heterogeneity of the nAChR is the reported coexistence of dispersed individual macromolecules with nanoclusters of variable sizes. From our previous camera-based SPT-STORM results at lower time resolution, we proposed that crowding of nAChRs in nm-sized aggregates (nanoclusters) may impede the motion of the individual receptor macromolecules[20,21]. We further proposed a picket-like mechanism to account for the transient motional hindrance. The experimental data could be accounted for in terms of a 2-state model[44] in which receptors switched between Brownian motion and obstructed diffusion (OD)[20,21]. The present results clearly show that the restricted diffusion sojourns also occur in the fast (low millisecond and sub-millisecond) time domain accessible to MINFLUX. Results from deep learning analyses[31] concurred with the early proposal. Using concatenated convolutional neural networks a more recent work

challenged the OD model against six other physical models; the 2-state model withstood the challenge and remains the simplest interpretation of the translational diffusion of the nAChR at the cell surface[45], a contention that appears to be extensive to the present experimental work.

The fluorescent cholesterol analogue fPEG-Chol employed in this study was chosen because (1) it has already been successfully used for labelling cholesterol-dependent domains in the same CHO-K1/A5 cell line employed in the present work[46]; (2) its polyethylene glycol chain maintains the fluorophore sufficiently far from the sterol moiety[47], making PEG-labelled lipids, and fPEG-Chol in particular, among the least biophysically and physiochemically membrane-perturbing probes[48,49]; this ensures functional preservation of indirect, membrane-mediated cholesterol effects on the membrane and direct cholesterol-nAChR interactions[50]; (3) its advantageous spectroscopic properties (with respect to absorption and emission wavelengths and photostability) for MINFLUX microscopy, and in particular for the 2-colour double-labelling scheme introduced in the present work, and (4) fPEG-Chol is known to reside almost exclusively at the cell-surface membrane, presumably at its outer, exoplasmic leaflet[46,47,51], with only a small fraction internalised by cells in the course of hours[49], well beyond the time window of our experiments.

As expected, the apparent diffusion of fPEG-Chol (alone or in the presence of the far-red CF®680R-BTX) was faster than that of the toxin-labelled nAChR protein (Table 1); the fluorescent cholesterol exhibited Brownian and superdiffusive and rarely subdiffusive motions (Fig. 1 and Table 2). MINFLUX further revealed that the motion of the nAChR, as assessed by its apparent anomalous diffusion constant $K_\beta$, was influenced by the presence of the fluorescent cholesterol probe (Table 1). While cholesterol depletion via methyl-β-cyclodextrin markedly slowed down nAChR diffusion, addition of 100 nM cholesterol only slightly accelerated diffusion (Supplementary Table 2), indicating that the membrane cholesterol concentration is close to saturating concentrations in CHO-K1/A5 cells, as previously reported[52]. Taken together, this series of experimental results strongly indicates that the 2D translational motions of cholesterol and nAChR are correlated.

## Cholesterol affects both the confined and non-confined portions of the nAChR SMTs

MINFLUX further afforded the spatiotemporal discrimination of the trajectories into their free diffusing segments and confinement sojourns, thus enabling calculation of the apparent diffusion coefficient in the two motional regimes. Whereas previous studies in cells and giant unilamellar vesicles yielded overall diffusion coefficient values of fPEG-Chol analogues close to ~1 μm² s⁻¹[51,53], here we could discriminate between a slow component in the confined sojourns ($1.10 \pm 0.01$ μm² s⁻¹) and the superdiffusive, 6-fold faster ($6.45 \pm 0.24$ μm² s⁻¹) component in the free walks (Supplementary Table 3). The same applies to the dissection of the diffusion coefficients of the BTX-labelled nAChRs, differing by a factor of 5 between free diffusing portions and confinement sojourns (Supplementary Table 3).

The duration of the non-confined portions of the BTX-labelled nAChR SMTs under control and cholesterol depletion conditions were in the order of ~15 ms (Fig. 2). These free-walking, random motion periods were statistically shorter in the presence of fPEG-Chol ($10 \pm 1$ ms, $p < 0.01$), i.e., falling beyond the range of those of the cholesterol probe alone ($9 \pm 1$ ms) or fPEG-Chol in the presence of CF®680R-BTX ($7 \pm 1$ ms). These experimental results further reinforce the notion of mutual dynamic interactions between cholesterol and the receptor protein. The effect of the neutral lipid was also manifested on the confined sojourns: cholesterol depletion increased by 2-fold the lifetime of the confined periods from $71 \pm 8$ ms (control) to $137 \pm 24$ ms (CDx) ($p < 0.01$). fPEG-Chol confinement sojourns lasted one order of magnitude less ($25 \pm 1$ ms) ($p < 0.0001$, Fig. 2), with no changes in $K_\beta$ in

the presence of the toxin. The same applies to $K_\beta$ in the non-confined portions of fPEG-Chol in the presence of the nAChR protein. However, confined portions became more subdiffusive (Fig. 2) when fPEG-Chol was sampled in the presence of CF®680R-BTX ($p < 0.001$). Exogenous cholesterol supplementation only reduced the duration of the confined sojourns of CF®680R-BTX ($p < 0.01$). Cholesterol also affected the co-tracking of the two molecules (Fig. 4). Undoubtedly, the main effect of cholesterol on nAChR diffusion is exerted on the confined portion of the nAChR trajectories. All in all, these kinetic parameters strongly suggest that the motion of nAChRs and cholesterol influence each other's diffusion behaviour.

## Trajectory directional (turning angle) and machine learning CONDOR analyses

nAChRs change the direction of their trajectories by step-by-step angular displacement. If the steps display no preference for moving forwards or backwards, as in Brownian motion, there is no turning angle predilection. If a molecule favourably adopts anticorrelated steps (i.e., the molecule tends to move backwards), as in confined diffusion, turning angles between 90° and 180° are characteristically observed, whereas when steps are positively correlated (i.e., the molecule tends to move forward), as in very fast, i.e., superdiffusive motions, they vary between 0° and 90°. Here we measured step lags instead of time lags due to the irregular time intervals obtained with MINFLUX recordings. We were not only able to validate previous camera-based STORM studies[20,21] but could also reveal that cholesterol accompanied the nAChR with a similar turning angle dependence. As expected, subdiffusive trajectories exhibited a strong preference for anticorrelated steps, a tendency that diminished as β increased (Supplementary Fig. 6).

When turning angle analysis was dissected into confined and non-confined portions, anticorrelated steps were found in confinement sojourns, whereas no preference was observed in non-confined, free-walk portions, which—as Fig. 3 points out—exhibited superdiffusive, Brownian, and subdiffusive behaviour. The results of turning angle analysis show that the model that best describes the heterogeneous behaviour of the nAChR is the two-state obstructed diffusion model[21]. Moreover, the strong step lag dependence of the probability density function suggested that structures of characteristic size were responsible for the obstructed diffusion[21,33], a feature disclosed with better precision using the MINFLUX measurement (Supplementary Fig. 4) and further confirmed by the deep learning CONDOR analysis (Supplementary Fig. 7 and Supplementary Table 5). Consequently, the fluorescent cholesterol analogue followed the same motional conduct as the nAChR.

## Overlap of nAChR and cholesterol single-molecule trajectories

One of the main findings of this work is the demonstration that fluorescent-labelled nAChR and cholesterol molecules can be co-tracked for distinct periods along their trajectories (Fig. 4). Using the overlap trajectory analysis outlined in "Methods", it was further possible to quantitatively estimate the spatial overlap of these joint motional periods, i.e., fPEG-Chol and CF®680R-BTX overlapped 25% and ~40% in confinement sojourns and non-confined portions, respectively (Fig. 5). Probabilistically, this represents a significantly high percentage ($p < 0.01$), considering the low concentration of CF®680R-BTX-labelled nAChRs (0.5 nM) and fPEG-Chol (1 nM) employed, the dimensions of the 2D host—the plasma membrane—the size and diffusion coefficients of the incumbent molecules, and the length of their trajectories. Upon addition of exogenous unlabelled cholesterol, the overlap was reduced to ~8% with 100 nM cholesterol in confined regions, probably reflecting a law of mass action-type displacement of the fluorescent cholesterol by the 100-fold higher concentration of unlabelled sterol. Similar effects were observed on the non-confined portions of the trajectories.

**The submembrane actin meshwork constitutes an additional hindrance to translational diffusion of nAChR and cholesterol**

The actin compartment is the submembrane meshwork or actin-based membrane skeleton fence ("MSK"), and thus very relevant to cell-surface molecules like the neurotransmitter receptor and its interacting cholesterol molecule. The notion of such a specialised actin supramolecular structure was introduced by Sheetz[54], who described the structure as "corrals" that restricted the lateral diffusion of membrane proteins, an idea subsequently examined in detail by Saxton and Jacobson[55,56]. Computer-enhanced high-speed SPT enabled the Kusumi group[57] to measure these corrals and extend the hypothesis to membrane lipids, leading in turn to the concept of time-dependent "hop diffusion": short-term confined diffusion within a compartment and infrequent, long-term hop diffusion between compartments of characteristic size[58], later supported by combined STED microscopy and fluorescence correlation spectroscopy (STED-FCS) data[59] and other SPT experiments based on, e.g., interferometric SCATtering or fluorescence microscopy[33]. In a previous STED superresolution imaging study, the cytoskeletal-disrupting drugs cytochalasin D and jasplakinolide were found to produce a statistically significant increase in the size of nAChR nanoclusters at the plasmalemma of CHO-K1/A5 cells, suggesting that the submembrane actin compartment affected the receptor distribution and local density at the cell surface[39].

CK-666 is a small molecule that stabilises the inactive state of Arp2/3, a protein complex involved in the nucleation of branched actin filaments that form the actin MSK corrals[36,37]. Treatment with CK-666 resulted in a modest increase in diffusion of both nAChR and fluorescent cholesterol, implicating the contribution of cytoskeletal fences or corrals[38] in addition to the pickets of clustered nAChR molecules that "self-restrict" the protein's autologous diffusion at the plasmalemma[20,21]. SPT measurements in different mammalian cells[59–61] showed that the actin MSK corrals have mesh sizes in the 40–100 nm range (specifically 40 nm in the case of CHO cells[60], the parental cell line of the CHO-K1/A5 clonal line used in the present work), in agreement with the current MINFLUX measurements showing apparent dimensions in the order of 50 nm. The 1.5-fold acceleration of the fast cholesterol diffusion upon CK-666-mediated actin disruption (Supplementary Table 2) agrees with literature values: fluorescent cholesterol is only modestly influenced by actin corrals[49,62]. While the diffusion regime remained unaltered, the percentage of the free diffusing component increased (Fig. 5), as did the anomalous diffusion constant $K_\beta$ (Table 2) and the distribution of joint co-tracking of nAChR and fluorescent cholesterol (Fig. 4). When analysed in terms of the hop-diffusion model using recently introduced criteria for MINFLUX studies[40], both the short-term intra-compartmental ($D_\mu$) and long-term inter-compartmental ($D_M$) diffusion coefficients increased upon CK-666 treatment (Fig. 6 and Table 3), suggesting disruption of corral fences. Interestingly, despite the increase in the diffusion coefficients of toxin-labelled nAChR and fPEG-Chol, the degree of overlap between the two molecules did not change, another manifestation of the interaction between the two within confinement sojourns and while crossing compartments. Altogether, the CK-666 experiments indicate that the actin submembrane cytoskeleton contributes to hindering the translational mobility of both receptor and cholesterol molecules, with only a minority (~10–20%) fraction of the motion-restricted molecules hopping from one compartment to another.

In conclusion, the observation of receptor and cholesterol trajectories with the enhanced spatiotemporal resolution and reduced photobleaching afforded by MINFLUX opens new possibilities to explore lipid-membrane protein interactions in live cells. The observation that the receptor and cholesterol trajectories overlap spatially and temporally is a finding that expands our knowledge of the mutual interactions between a neurotransmitter receptor protein and the neutral lipid at the plasma membrane[50,63]. Achieving sub-millisecond time resolution and spatial resolving power of ~7 nm, the MINFLUX study surpasses the capabilities of traditional camera-based techniques like STORM, unveiling individual step characteristics, dissecting the lifetimes of the free walks and confinement sojourns, and several other metrics of the single-molecule trajectories in a hitherto unexplored time domain.

Man-tailored depletion/enrichment of cholesterol content confirms the interplay between the neutral lipid and nAChR dynamics, and CK-666 inhibition of Arp2/3 actin nucleation provides evidence of the additional, albeit minor, contribution of the submembrane cortical actin meshwork in restricting the translational motion of the nAChR and cholesterol. The directionality (turning angle) analysis of the SMTs revealed distinct preferences for step correlations, especially within confined sojourns, where steps were markedly anticorrelated in the subdiffusive sojourns. The ability to dissect free walks from confined sojourns facilitated the recognition of overlaps of nAChR and cholesterol trajectories in confined and non-confined areas, thus providing a strong piece of evidence of the tight relationship and mutual interactions between cholesterol and nicotinic receptor dynamics.

## Methods

### Materials

Fluorescent polyethylene glycol K114-labelled cholesterol (fPEG-Chol, trademarked as AbberiorStar Red-PEG-Chol) was purchased from Abberior GmbH, Göttingen, Germany. A stock solution (1 mg/mL) in ethanol was kept at −20 °C. CF®640R-labelled α-bungarotoxin (CF®640R-BTX excitation/emission max.: 642/663 nm) and CF®680R-labelled α-bungarotoxin (CF®680 R, excitation/emission max.: 680/701 nm), were purchased from Biotrend GmbH, Köln, Germany. Methyl-β-cyclodextrin (product No. 332615), cholesterol-methyl-β-cyclodextrin complex (water-soluble cholesterol, product No. C4951) and the Arp2/3 inhibitor I CK-666 (product No. 182 515) were purchased from Sigma-Aldrich. Colloidal gold nanoparticle fiducials (150 nm) were purchased from BBI Solutions, Crumlin, U.K.

### Sample preparation

CHO-K1/A5 cells were grown as originally reported[29]. Following a washing step with cell culture medium without phenol red, cells were incubated for 5 min on ice with a blocking medium consisting of 1 mg/mL bovine serum albumin in PBS, followed by 30 min labelling on ice with either CF®640R-BTX, CF®680R-BTX, fPEG-Chol or a combination of CF®680R-BTX + fPEG-Chol in 1 mL of culture medium. After several initial assays, the final concentration of the two fluorescent α-bungarotoxin derivatives was kept at 500 pM, and that of the fluorescent cholesterol analogue at 1 nM. Gold fiducials were added to the sample chamber containing the coverslip-adhered cells and allowed to settle while the sample was mounted in the microscope stage.

CDx treatment was performed as previously reported[52] by incubation of the cells with 10 mM CDx in PBS for 20 min on ice before incubation with the fluorescent dyes. Cells were treated with 10 mM CK-666 in PBS for 15 min at room temperature. Soluble cholesterol was added at two concentrations (50 and 100 nM).

### MINFLUX experiments

Measurements were performed on a MINFLUX setup from Abberior Instruments GmbH (Göttingen, Germany), based on the iterative localisation approach[42]. To localise the emitter and collect emitted photons, we used a hexagonal-shaped beam pattern of diameter $L$ (see Supplementary Table 6) with points equidistant to the beam centre, which features zero intensity. The use of a circularly polarised doughnut-shaped zero-intensity beam pattern provided an orientation-independent localisation of the fluorophore emission dipoles. The position of the emitter was refined through an iterative sequence of consecutive pattern scanning operations that zoomed in onto the emitter with decreasing pattern diameter ($L$) values, as shown in Supplementary Table 6. These sequences are performed

automatically by the microscope control electronics. In short, MIN-FLUX repeats the first iteration until a fluorescence signal is detected, aligns the zero-intensity centre of the excitation beam, which is precisely known, with the position of the emitter, at which point the microscope will seek to improve the localisation by performing successive iterations (4 in the present work) with an increasingly small beam diameter, thus bringing the doughnut beam zero centre closer to the emitter at each iteration. Tracking is then achieved by repeating the last iteration until the signal is lost, at which point the process starts anew. Other essential parameters for the 2D scanning sequence, provided by the manufacturer, are listed in Supplementary Table 6. Another variable that was modified between pattern iterations is the excitation laser power. This was increased in discrete steps between iterations with the laser power multiplier parameter listed in the table, which refers to a reference excitation power of 1.78 μW at the exit of the objective. The sampling rate of the tracking measurements performed with this method was between 5 and 10 kHz. The localisation precision of the experiments was calculated from the non-linear fitting of the MSD curves (explained in two sections further on). We obtained $\sigma = 7 \pm 1$ nm. All ROIs were manually identified using the MATLAB roipoly function implemented in Python: https://github.com/jdoepfert/roipoly.py.

### Initial analysis of MINFLUX experimental series and immobile molecule exclusion

Tracks with irregular time intervals were analysed using object- and array-oriented routines written in Python. A few tracks consisting of only one step ("frustrated" events) were detected and excluded from further analysis. Each complete dataset, consisting of several thousand steps, was analysed, including all validated trajectories; the same set of data was next subjected to the criterion of ref. 30 to exclude immobile particles from the single-molecule tracking (SMT) analysis as in refs. 20,21. The above criterion is based on the ratios of the radius of gyration $R^2_g$, defined as:

$$R^2{}_g = \frac{1}{N} \sum_{i=1}^{N} \left( \vec{r}_i - \langle \vec{r} \rangle \right)^2 \qquad (1)$$

where $\vec{r}_i$ is the coordinate at time step $i$ out of $N$ total time steps and the mean step size $\langle |\Delta r| \rangle$ of the recorded trajectories. Trajectories classified as immobile using the aforementioned criterion were not analysed further.

In the case of the fPEG-Chol alone and fPEG-Chol (+CF®640R-BTX), the complete raw datasets were excluded from threshold selection, because the distribution of the normalised ratios in these datasets is apparently uniform. Including these ratios into the threshold selection (which depends on the 95% confidence intervals of the mean) would select an overestimated value, leading to incorrectly classifying trajectories as immobile. Supplementary Fig. 3 shows the histogram of the normalised ratios and examples of trajectories classified into immobile and mobile.

### Discrimination between fPEG-Chol and CF®680R-BTX in double-labelling experiments

In the case of samples co-labelled with both fPEG-Chol and CF®680R-BTX, to discriminate between the two signals, we used a Detector Channel Ratio (DCR) criterion, as schematically shown in Supplementary Fig. 9. The DCR criterion was applied to all individual points of the trajectories. If the average DCR of a trajectory was above a certain threshold, the particle was assigned to fPEG-Chol tracks. Otherwise, the track was assumed to correspond to CF®680R-BTX. To establish the threshold, we plotted histograms of the DCR when both probes were recorded separately, as shown in Supplementary Fig. 2. A difference between the two mean distributions was observed, and a

threshold of -0.55 was applied as it discriminated well between the two distributions.

### Turning angle analysis

Turning angle is conventionally measured for increasing time lags $\Delta$[33,64]. As the intervals in the MINFLUX series are not regular, step lags were used instead of time lags. Hence, the relative angle $\theta(i;\Delta)$ distended between successive steps is defined as:

$$\cos[\theta(i;\Delta)] = \frac{\mathbf{V}(i;\Delta) \cdot \mathbf{V}(i+\Delta;\Delta)}{|\mathbf{V}(i;\Delta)| |\mathbf{V}(i+\Delta;\Delta)|} \qquad (2)$$

such that $\mathbf{V}(i;\Delta) = \vec{x'}_{i+\Delta} - \vec{x}_i$. Basically, $\theta(i;\Delta)$ is the angle between $\mathbf{V}(i;\Delta)$ and $\mathbf{V}(i+\Delta)$. The formula was iterated through all trajectory positions whenever possible. The probability density function (PDF) for increasing step lags was obtained from the histogram with a bin width = 10. The algorithm described in ref. 64 was vectorised to improve computing times.

### Analysis of trajectory kinetic parameters

For a given trajectory $j$ and number of $N_{m\Delta_t}$ displacement intervals between $\Delta_O + (m\text{-}1)\,\Delta_t$ and $\Delta_O + m\Delta_t$, the Time-Averaged MSD (TA-MSD) is defined as:

$$TA - MSD\left(t_{lag} = m\Delta_t\right) = \left\langle \Delta r^2 \left(t_{lag} = m\Delta_t\right) \right\rangle_T = \frac{1}{N_{m\Delta_t}} \sum_{i=1}^{N_{m\Delta_t}} d_j(t_i, t_{i'})^2 \qquad (3)$$

$$d_j(t_1, t_2) = [x_j(t_2) - x_j(t_1)]^2 + [y_j(t_2) - y_j(t_1)]^2 \qquad (4)$$

Where $x_j$ and $y_j$ are the coordinates of the trajectory on the 'x' and 'y' axes, $\Delta_0 + (m-1)\Delta_t < t_{i'} - t_i < \Delta_0 + m\Delta_t = 132$ μs, $\Delta_O = 84$ μs and $i$ the interval number. To calculate the TA-MSD, we modified the Python implementation of ref. 65 to be suitable for irregular intervals. The only difference of this expression from the one presented in ref. 40 is that displacements are binned into intervals of width $\Delta_t$ starting in $\Delta_0$ due to the variable sampling rate in MINFLUX setup of this work. The TA-MSD was fitted to the function:

$$\left\langle \Delta r^2 \left(t_{lag} = m\Delta_t\right) \right\rangle_T = \Gamma * (m\Delta_t)^{\beta-1} * 2 * n_d * \Delta_t * m * \left(1 - \frac{2R}{m}\right) + 2 * n_d * \sigma^2 \qquad (5)$$

where $R$ is a factor that accounts for the motion blur caused by molecular motion during the time of the measurement, $\sigma$ is the dynamic localisation uncertainty, and $n_d$ is the number of dimensions in which the diffusion takes place. Here, for molecules diffusing in the plane of the membrane, $n_d = 2$ for 2D trajectories. In addition, we set $R = 1/6.2$[10], which assumes homogeneous illumination during the experiments[40,41].

The anomalous exponent ($\beta$), the transport coefficient ($\Gamma$) and $\sigma$ were obtained by fitting MSD points up to 50 ms step points of the individual time-averaged MSDs (TA-MSD). If $R = 0$ and $\Delta = 0$, we found that, compared to the equation $\left\langle \Delta r^2 \left(t_{lag} = m\Delta_t\right) \right\rangle_T = K_\beta t^{\beta}$[21], we obtained $K_\beta = \Gamma * 2 * n_d$. Trajectories were separated into subdiffusive ($\beta \leq 0.9$), Brownian ($0.9 < \beta \leq 1.1$) and superdiffusive ($\beta > 1.1$). This classification criterion is a simplified version of the one outlined in refs. 20,21. Finally, all fittings with a residual standard deviation greater than 1000 nm² or MSD curves with ≤50% of available points were rejected.

### Tracking recurrence analysis

Trajectories were found to be interrupted by periods of confinement (i.e., a nAChR particle repeatedly "visited" the same area). Quantitative metrics of the confinement sojourns were obtained using the algorithm of Krapf and coworkers[66]. Such algorithm "draws" a circular area

between consecutive localisations and measures the number of times that the particle calls on the given area. If the particle visited the area more than $V_{th}$ times, the particle was considered transiently confined. Initially, this threshold, which depends on the experimental conditions, was set to a value of 11, as used for experimental data in ref. 66. This threshold erroneously detected confinement in places where, through visual inspection, the moving particle appeared to follow a Brownian behaviour. $V_{th}$ was therefore increased to a value = 33. To prevent false state transitions within a trajectory, we took the sequence of states assigned by the algorithm and divided it into non-overlapping windows of 3 steps. Next, each step within the window was reassigned to the state that occurred most frequently within that specific window. To analyse confinement sojourn parameters, trajectories were segmented into sub-trajectories between confined and mobile states. The shape of the confinement area was fitted using an algorithm that determines the eccentricity and length of the semi-axes defined by an ellipse fitted on these data points. The area of confinement zones is equal to the area defined by their convex hull.

### Spatial overlap in confined and non-confined regions of the trajectories

In cells co-labelled with fPEG-Chol and the nAChR fluorescent antagonist CF®680R-BTX, the trajectories of the two probes frequently intersected. To determine whether these intersections occurred in confined or non-confined portions of the trajectories and how recurrent this phenomenon was: (1) the confinement areas of each CF®680R-BTX trajectory ($C_{CF®680R-BTX}$) were considered to overlap with the confinement areas of fPEG-Chol trajectories ($C_{CF®680R-BTX \cap fPEG-Chol}$) if their convex hulls intersected. An apparent overlap coefficient, $C = C_{CF®680R-BTX \cap fPEG-Chol} / C_{CF®680R-BTX}$ was also calculated for each trajectory. This metric can also be defined as the ratio $C = C_{fPEG-Chol \cap CF®680R-BTX} / C_{fPEG-Chol}$; (2) to characterise the overlap in non-confined, free-walk portions of the trajectories, the spatial proximity between the individual steps of fPEG-Chol and CF®680R-BTX was estimated using k-d tree structures[67]. A k-d tree is a multidimensional tree-like structure that enables efficient spatial queries, such as the proximity between trees. The k-d tree implementation included in the SciPy software package[68] and the polygon intersection algorithm from Shapely[69] were used to this end. Steps from two distinct k-d trees are considered close if they are within a distance $<r$. This parameter was estimated for each CF®680R-BTX and fPEG-Chol non-confined portions by querying their corresponding k-d trees using a value of $r = 10$ nm. Steps were considered to overlap if they were within a distance smaller than this value, equivalent to the localisation precision.

### Spatial confinement overlap significance test

Measuring confinement overlaps using the convex hull was found to be too slow for estimating the statistical significance of the overlap between confinement portions. Instead, we resorted to the Intersection over Union (IoU) metric in the following manner. First, the ROI was divided into square cells of 7 x 7 nm. The ROIs were then converted into two matrices: one for fPEG-Chol and another for CF®680R-BTX. The fPEG-Chol matrix was assigned a value of 1 if there was at least one fPEG-Chol confined localisation in that cell. The IoU was defined as the intersection (cells with both fPEG-Chol and CF®680R-BTX) divided by the union (cells with either fPEG-Chol or CF®680R-BTX) of the two matrices. To determine if the IoU was significant, we used the Andi-dataset package[70] to simulate trajectories within an ROI of the same dimensions. First, we defined the dimensions of the ROIs where the trajectories are located. Next, for each actual experimental trajectory, a simulated trajectory was placed within the ROI, ensuring it moved through confinement zones randomly placed within the ROI. These zones had the same radius as those obtained previously using the recurrent algorithm, covering ~10% of the ROI area. This process was repeated for CF®680R-BTX trajectories. The IoU of the simulated

experiment was measured next. The resulting calculated IoUs were then compared with the experimental IoU values using the percentile $\alpha' = 1 - \alpha$ percentile (significance level of $\alpha = 0.05$) of the randomly simulated IoUs. If the actual IoU was larger than the $\alpha'$ percentile, the overlap was significantly higher than that expected by chance. The same procedure was applied to non-confined portions. This analysis ensured that the significance of the overlap was assessed rigorously through statistical testing, providing robust conclusions about the spatial relationships between the two probe channels in the ROI.

### Particle dynamics characterisation between free and hop diffusion

To categorise the diffusional behaviour between free and hop diffusion, we followed the workflow described in ref. 40, which is based on ref. 41. Trajectories were analysed by fitting their MSD curves to two different models: $MSD_{free}$ and $MSD_{hop}$. Prior to the analysis proper, all trajectories were truncated to segments of 500 localisations each to ensure uniform statistical sampling for each trajectory.

If the particles diffuse freely in 2D, the MSD is described as:

$$MSD_{free}(n\Delta_t) = 2n_d D n\Delta_t \tag{6}$$

$MSD_{hop}$ combines $MSD_{free}$ and $MSD_{conf}$ (i.e., the MSD describing a spatially confined particle). $MSD_{hop.}$ is described as:

$$MSD_{hop} = MSD_{free}(n\Delta_t) + \frac{D_\mu - D_M}{D_\mu} MSD_{conf}(n\Delta_t) \tag{7}$$

where $D_\mu$ and $D_M$ are the short-term (*intra-compartmental*) and long-term (*inter-compartmental*) diffusion coefficients, respectively (Rickert, 2024). Analytically speaking, if the molecule diffuses freely, $D_M = D_\mu \left( \frac{D_\mu - D_M}{D_\mu} = 0 \right)$; if the molecule is totally confined within a single confinement region along its lifetime, $D_M = 0$[41].

The type of diffusion described by $MSD_{hop}$ occurs when confinement regions are partially permeable, and the molecules have a probability $p_{hop} > 0$ to jump to another confinement region. Following the approaches of ref. 71 and ref. 40, $MSD_{conf}$ can be formulated as:

$$MSD_{conf}(n\Delta_t) = A \left[ 1 - \exp\left( -\frac{n\Delta_t}{\tau} \right) \right] \tag{8}$$

where $A$ is the average distance between two randomly chosen points within the confinement regions and $\tau$ is the equilibration time at which the effect of boundaries appears and the MSD reaches its plateau region[6]. Since Hell and coworkers defined the expressions for 3D trajectories in a 3D space, and hence the confinement regions have a volume[40], the expressions had to be adapted to two dimensions. Confined diffusion can be restricted to a square domain of side $L$, and hence $A = L^2/3$, where $L$ is the apparent confinement domain size[72]. The equilibration time, $\tau = A/(4 D_\mu)$, such that $D_\mu$ is the diffusion coefficient that dominates the diffusion within the confinement region[71], results in:

$$\tau = \frac{L^2}{12 D_\mu} \tag{9}$$

A squared confined domain $L$ is taken for the sake of simplicity, but any 2D geometric shape can be considered[72] in the $MSD_{conf}$:

$$MSD_{conf}(n\Delta_t) = \frac{L^2}{3} \left[ 1 - \exp\left( -\frac{12 D_\mu n\Delta_t}{L^2} \right) \right] \tag{10}$$

Having obtained the expressions for $\text{MSD}_{conf}$ and $\text{MSD}_{free}$, $\text{MSD}_{hop}$ can be rewritten for 2D tracks as:

$$
\begin{aligned}
\text{MSD}_{hop}(n\Delta_t) &= \text{MSD}_{free}(n\Delta_t) + \frac{D_\mu - D_M}{D_\mu}\text{MSD}_{conf}(n\Delta_t) \\
&= 2n_d D_M n\Delta_t + \frac{D_\mu - D_M}{D_\mu}\frac{L_{hop}^2}{3}\left[1 - \exp\left(-\frac{12D_\mu n\Delta_t}{L_{hop}^2}\right)\right] \\
&= 2n_d n\Delta_t\left\{D_M + \frac{D_\mu - D_M}{D_\mu}\frac{L_{hop}^2}{6n_d n\Delta_t}\left[1 - \exp\left(-\frac{12D_\mu n\Delta_t}{L_{hop}^2}\right)\right]\right\}
\end{aligned}
\tag{11}
$$

such that diffusion coefficients $D_\mu$ and $D_M$ are separated by a length scale $L_{hop}$[40].

Application of this analysis returned in some cases a $D_M \approx 0$. For this reason, we decided to include $\text{MSD}_{conf}$ within the scope of the analysis.

Thus far, $\text{MSD}_{hop}$ and $\text{MSD}_{free}$ do not include corrections for measurement artefacts. To account for these, $\text{MSD}_{hop}$ and $\text{MSD}_{free}$ were rewritten[40] as:

$$
\text{MSD}_{free}(n\Delta_t) = 2n_d D\Delta_t(n - 2R) + 2n_d\sigma^2
\tag{12}
$$

$$
\text{MSD}_{hop} = 2n_d\Delta_t\left\{D_M + \frac{D_\mu - D_M}{D_\mu}\frac{L_{hop}^2}{6n_d n\Delta_t}\left[1 - \exp\left(-\frac{12D_\mu n\Delta_t}{L_{hop}^2}\right)\right]\right\}(n - 2R) + 2n_d\sigma^2
\tag{13}
$$

Because $\text{MSD}_{conf}$ is equal to $\text{MSD}_{hop}$ when $D_M = 0$, $\text{MSD}_{conf}$ is rewritten as:

$$
\text{MSD}_{conf} = \frac{L^2}{3}\left[1 - \exp\left(-\frac{12D_\mu n\Delta_t}{L^2}\right)\right](n - 2R) + 2n_d\sigma^2
\tag{14}
$$

In the case of $\text{MSD}_{free}$, the free parameters are $D$ and $\sigma$. In the case of $\text{MSD}_{hop}$, the free parameters are $D_M$, $D_\mu$, $L_{hop}$, and $\sigma$. Finally, in the case of $\text{MSD}_{conf}$, the free parameters are $D_\mu$, $L_{hop}$, and $\sigma$. $\text{MSD}_{free}$ derived in Eq. 12 has a different nomenclature but is essentially the same expression employed by Hell and coworkers[10].

To classify trajectories, the following steps were followed: (1) an MSD curve was calculated for each trajectory; (2) the initial 20% data points of the MSD curves were fitted to $\text{MSD}_{free}$, $\text{MSD}_{hop}$, and $\text{MSD}_{conf}$ solving three non-linear optimisation problems; the free parameters were adjusted such that the sum of the squared residuals (SSR) of both regressions on the observed data were minimised. We used the Python-based software package SciPy to solve this optimisation problem 100 times (the solution with the least SSR was picked for each case). MSD points were sampled logarithmically to emphasise the fitting to the initial 20% MSD points. (3) Next, the Bayesian Information Criterion (BIC) was calculated for both fits:

$$
\text{BIC} = n * ln\left(\frac{\text{SSR}}{2}\right) + k * ln(n)
\tag{15}
$$

where $n$ and $k$ are the number of data points and number of fit parameters, respectively.

In Rickert et al.[40], the MSD curves are fitted to $\text{MSD}_{hop}$ with constraint $D_\mu > 5\,D_M$ to find substantial differences between the two diffusion coefficients. However, this constraint artifactually imposes a minimum value of 5 for $S_{conf} = D_\mu / D_M$, which is defined as the confinement strength (i.e., the normalised residence time of a confined molecule) and $S_{conf} > 1$ (Wieser, 2015), and hence this constraint was removed, leaving $D_\mu > 5\,D_M$. Supplementary Table 7 depicts the boundaries of the initial free parameter values for each MSD model to solve the optimisation problem.

## Statistics and reproducibility

All graphs and statistics were prepared following the guidelines of ref. 73, where it is suggested that the average of ROIs be taken as single reported values to increase statistical robustness. All reported mean values and statistical tests were conducted on ROI averages. To compare distributions, we used the Kolmogorov–Smirnov (KS) test for two samples (null hypothesis was rejected for $p$-value < 0.05). To compare three or more distributions, the Kruskal–Wallis test was used. Statistical analysis was carried out using GraphPad Prism 8.

## Reporting summary

Further information on research design is available in the Nature Portfolio Reporting Summary linked to this article.

## Data availability

All data gathered for this study are available upon request to the corresponding authors. Upload of the data on a public data server has so far not been conducted due to the sheer amount of data. Source data are provided with this paper.

## Code availability

Python code for the analysis of the data is available at the public repository Zenodo [https://doi.org/10.5281/zenodo.15389696] and on GitHub [https://github.com/lucasSaavedra123/minflux_analysis].

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

## Acknowledgements

We thank Alejo Mosqueira and Bela Tristan Leander Vogler for comments and suggestions on the manuscript. C.E. and F.J.B. acknowledge the Alexander von Humboldt Foundation for the Research Group Linkage Programme grant to the laboratories in Jena and Buenos Aires, respectively, and the Deutsche Forschungsgemeinschaft (DFG, Instrument funding MINFLUX Jena INST 275_405_1) for funding of the MINFLUX microscope. We also gratefully acknowledge financial support from the Deutsche Forschungsgemeinschaft (DFG, German Research Foundation; Germany's Excellence Strategy—EXC 2051—Project-ID 390713860; project number 316213987—SFB 1278; GRK M-M-M: GRK 2723/1—2023—ID 44711651; Instrument funding modular STED INST 1757/25-1 FUGG), the State of Thuringia (TMWWDG), the Leibniz Association (Leibniz Science Campus InfectoOptics Jena financed by the funding line Strategic Networking of the Leibniz Association, project number W8/2018), and the Free State of Thuringia (TAB; Advanced STED /FGZ: 2018 FGI 0022; Advanced Flu-Spec / 2020 FGZ: FGI 0031; Multi-XUV / 2023 FGR 0054).

Further, this work is supported by the Photonics Research Germany (FKZ: 13N15713 / 13N15717) and is integrated into the Leibniz Centre for Photonics in Infection Research (LPI). The LPI initiated by Leibniz-IPHT, Leibniz-HKI, UKJ and FSU Jena is part of the BMBF national roadmap for research infrastructures. A part of the project on which these results are based was funded by the Free State of Thuringia under the number 2018 IZN 0002 (Thimedop) and co-financed by funds from the European Union within the framework of the European Regional Development Fund (EFRE).

## Author contributions

F.J.B. and C.E. conceived the project; F.R. and F.J.B. performed the live-cell experiments; L.A.S. wrote the analysis code. L.A.S. and F.J.B. analysed the data; F.J.B. wrote the manuscript with feedback from the authors; F.J.B. edited the manuscript and wrote the replies to the reviewers.

## Funding

## Competing interests

The authors declare no competing interests.
