## [Transparent Peer Review file · Nature Communications]

Concurrent diffusion of nicotinic acetylcholine receptors and fluorescent cholesterol disclosed by two-colour sub-millisecond MINFLUX-based single-molecule tracking

Corresponding Author: Professor Christian Eggeling

Version 0:

Reviewer comments:

Reviewer #1

(Remarks to the Author)

The authors here present a very nice paper with original data and novel learning about the impact of cholesterol on protein tracking. We recommend the paper is published.

The authors simultaneously investigated the dynamics of nicotinic acetylcholine receptors and cholesterol in the plasma membrane of living cells. Overall, this manuscript presents an interesting exploration of the combination of the minimal photon flux (MINFLUX) microscopy technique with data analysis methods to study membrane protein dynamics and membrane lipids on the cell surface.

The manuscript is well-written, and the experimental approach using the MINFLUX technique for co-tracking biomolecules in biological membranes is novel and relevant in the context of understanding synaptic transmission. However, some details still need to be addressed, as outlined below

Minor concerns

MINFLUX microscopy relies on a tightly focused laser beam to achieve nanometric precision in single-molecule tracking. However, this intense laser focus can lead to local heating effects, that may alter membrane fluidity and protein dynamics.

1- Could the authors provide further clarification on the experimental design and controls implemented to minimize these artifacts, ensuring accurate and biologically relevant measurements?

2- To better understand the distribution of the nAChR receptor fused to CF640-BTX and the Cho molecules (fPEG-cholesterol) in the plasma membrane, we suggest that the authors include representative images of cells with the double-stained in supplementary material. As mentioned on page 5, the authors inspected at least 10 cells for each experimental condition.

3- It is suggested to include in the text, at line 106, the excitation wavelengths for both probes, as well as the emission wavelengths that allow discrimination between them by ratiometric measurement.

4- Review in the text of the supplementary material, line 12 "A dye which emits only in I1 will have a DCR value of 1, and its DCR will be 0." Considering the formula used to calculate the DCR value, if a molecule emits only in I1, its DCR will be 1 and its I2 would be 0.

5- In Supplementary Figure 1, in the DCR histograms of experiments with simultaneous staining using CF@680R-BTX and fPEG-Chol (right), a difference between the two mean distributions can be observed, more markedly in the distribution of fPEG-Chol alone and when co-labeled with CF@680R-BTX. Additionally, a broader distribution can be observed in the histograms, indicating the contribution of CF@680R-BTX emission signals to fPEG-Chol emission and vice versa. Could you briefly explain the reason for these seemingly significant differences and whether you considered any correction factor to account for false positives in order to isolate trajectories with fPEG-Chol and BTX680R when both experimental labeling conditions were sampled simultaneously?

6- In Table 1, in the section Perturbation of the submembrane actin cytoskeleton network by CK-666, Far-red toxin (CF@680R-BTX)-labelled nAChR upon CK-666 treatment results are not found (Far-red toxin (CF@680R-BTX)-labelled nAChR upon CK-666 treatment). Review and ensure that the nomenclature is consistent with the rest of the table.

7- In line 202, If you are referring to other studies where SMT analysis has reported β threshold values higher than those in this work for trajectory classification, it missing the references.

8- Previous studies have reported the lateral diffusion of transmembrane receptors, demonstrating heterogeneous diffusion

behavior. It has also been proposed that interactions with the cytoskeletal mesh or lipid domains could constrain membrane receptor movement in the plasma membrane (PM) using various fluorescence microscopy techniques. In this work, the study by Mosqueira et al. (2018, 2020) has been cited, reporting that the translational motion heterogeneity of the nAChR is characterized by the coexistence of dispersed individual macromolecules and nanoclusters of variable sizes. Considering that these nanoclusters or nanodomains are highly dynamic structures, a broader discussion based on the results of this study would be relevant.

9- Correct the definition of (PDF) in Figure 3. If possible, simplify Figure 3, as it contains a large amount of information.

Consider presenting only the probability density functions corresponding to fewer step lags.

10- Introduce the full meaning of the acronym MINFLUX the first time it appears in the text.

Reviewer #2

(Remarks to the Author)

Dear Editor,

The manuscript entitled “Concurrent diffusion of nicotinic acetylcholine receptors and fluorescent cholesterol disclosed by two-colour sub-millisecond MINFLUX-based single-molecule tracking” by Reina et al., presents a study of a neurotransmitter receptor at the plasma membrane of mammalian cells, using a modern and high-end technique (MINFLUX microscopy). The experiments are well performed and explained in detail, the conclusions are properly supported by the results, and the manuscript is in general well written. Considering the fast development of these kind of techniques, providing a higher spatial and temporal resolution, this study should appeal a broad reader community. However, some concerns must be addressed, before the manuscript can be recommended for publication:

The authors have vast experience in this system, and have already presented studies using other established techniques such as SPT-STORM, FCS, FRAP (e.g. Eur Biophys J (2010) 39:213–227, Almarza 2014, Mosqueira 2018). Thus, in my opinion, the main novelty of the manuscript is the successful application of MINFLUX tracking to this system for the first time. As this technique offers a higher spatial and temporal resolution, this provides the possibilities to reveal phenomena otherwise hidden in other classical techniques. The manuscript could be improved by using the Discussion section to describe the advantages observed (not of the technique itself, but the advantages observed in practice) and for a close comparison with results previously obtained with methods of lower spatiotemporal resolution. In particular, if there are some new phenomena that was not observed before, if there are experiments or data analysis performed for the first time here, that were not possible before due to limitations of classical techniques, and which results and measured parameters agree or not with previous reports, or are reported here with a higher accuracy. The reader should be able to assess this information without the need to reach for a larger number of references.

Other concerns:

- 1) Did previous studies (from this or other authors) performed two-color SPT, FCS (or any other technique) on the same system? If this is the case, the results should be also compared, and if not, the novelty of this work regarding this aspect should be highlighted. As it reads, it is clear this is the first study presenting two-color MINFLUX tracking using spectral discrimination, but it is not clear if this is the first co-tracking experiment of this receptor by any method.
- 2) The dyes used are referred as CF640/CF680 and later CF640R/CF680R, but these are different dyes. According to the vendor CF680 is a cyanine-based dye and CF680R a rhodamine-based dye, both with quite different properties. CF680 has been used for MINFLUX imaging, in part due to its tendency to blink, which is undesired for tracking. The actual dyes used should be clarify, and eventually any issues with blinking discussed.
- 3) Intuitively, the probes are added at a very low concentration and should not affect or modify the system, or at least have a minimal perturbation. However, results differ when a different probe is used: for instance, CF640R-BTX and CG680R-BTX results in table 1, or “the motion of the nAChR, as assessed by its apparent anomalous diffusion constant $K\beta$, was influenced by the presence of the fluorescent cholesterol probe” (page 25). Was there any attempt to lower the concentration of the probes?
- 4) Regarding the quantitative analysis of overlap between labelled nAChR and Chol trajectories: do the results depend on the concentration of the fluorescent probes used for labelling? Considering the low concentration of the probes, a 70 % trace overlap seems too high. Is there an estimation of the percentage of nAChR receptors and Cholesterol molecules labelled? Are all receptors labelled?
- 5) In the colour discrimination (DCR analysis), a threshold of 0.55 is defined in Supplementary Figure 1 and in figure 4. However, in the main text (pages 15-16), the analysis is based on a double threshold of 0.4 and 0.55, with this central window used for colocalization. This should be clarified.

With best regards

Reviewer #3

(Remarks to the Author)

Version 1:

Reviewer comments:

Reviewer #1

(Remarks to the Author)

All concerns have been addressed and I recommend the paper published. It is a very nice paper.

Reviewer #2

(Remarks to the Author)

Dear Editor,

In the revision of the manuscript entitled "Concurrent diffusion of nicotinic acetylcholine receptors and fluorescent cholesterol disclosed by two-colour sub-millisecond MINFLUX-based single-molecule tracking" by Reina et al., the authors have addressed all concerns from the original submission. The manuscript has been considerably improved, and therefore, I can recommend it for publication.

Reviewer #3

(Remarks to the Author)

First, we would like to thank the reviewers for their constructive and positive comments, which have helped us to improve the quality of the work. We have re-drawn several figures in both the main text and Suppl. Material, incorporated a new Figure in the Material and Methods section as well as two in the Supplementary Material.

Please find enclosed our point-by-point responses, marked in bold.

We include colour-tracked change versions of the manuscript and Supplementary Material, in addition to clean copies of the same.

Replies to reviewers' comments

Reviewer #1 (Remarks to the Author):

The authors here present a very nice paper with original data and novel learning about the impact of cholesterol on protein tracking. We recommend the paper is published.

The authors simultaneously investigated the dynamics of nicotinic acetylcholine receptors and cholesterol in the plasma membrane of living cells. Overall, this manuscript presents an interesting exploration of the combination of the minimal photon flux (MINFLUX) microscopy technique with data analysis methods to study membrane protein dynamics and membrane lipids on the cell surface.

The manuscript is well-written, and the experimental approach using the MNFLUX technique for co-tracking biomolecules in biological membranes is novel and relevant in the context of understanding synaptic transmission. However, some details still need to be addressed, as outlined below

Reply: We thank the reviewer for the positive comments and constructive suggestions. We have addressed them all, as replied below.

Minor concerns
MINFLUX microscopy relies on a tightly focused laser beam to achieve nanometric precision in single-molecule tracking. However, this intense laser focus can lead to local heating effects, that may alter membrane fluidity and protein dynamics.
1- Could the authors provide further clarification on the experimental design and controls implemented to minimize these artifacts, ensuring accurate and biologically relevant measurements?

Reply: This is a valid and very important comment; we have dealt with possible light-induced deleterious effects, either directly addressing photobleaching or other photo-induced effects or introducing appropriate controls and remedies (see references below). It follows from these insights that we can safely neglect the possible influence of light-induced artifacts in the present work. In line with the reviewer's comment, we have added the following paragraph in the main text and Supplementary Material addressing this important point:

Main text (page 6): *Our experimental conditions minimize photo-induced biasing effects such as unwanted loss or arise of signal, slow-down or speed-up of diffusion, and cell death due to photobleaching, photoblueing, photoblinking, phototoxicity, or local heating and trapping, as highlighted in detail in the Supplementary Material.*

New section added to Suppl. Material in the revised manuscript:

Dismissal of possible biasing effects resulting from laser illumination

Fluorescence experiments are generally exposed to biasing effects from the laser illumination. Main possible light-induced effects are photobleaching, photoblinking or photoblueing of the labels (leading to loss of signal or arise of unwanted more blueish emission), phototoxic effects on cells (leading to cellular changes and death), photothermal effects (eventually heating up the membrane and thus speeding up diffusion), and laser-based trapping of objects (leading to artificial trapping of molecules). The conditions producing such effects with deleterious changes in membrane fluidity and protein dynamics are not met by the present experimental conditions using Minflux in single-particle tracking mode, which is particularly advantageous in this regard, since: 1) focal laser intensities are one or even more orders of magnitude below those used in confocal or STED microscopy measurements, reducing detrimental effects by photobleaching, photoblinking and photo-blueing (Eggeling et al. 1998; Eggeling et al. 2005; Dasgupta et al. 2024); 2) only one excitation wavelength was used in the two-colour excitation, excluding the possibility of multi-step absorption events (Eggeling et al. 2006; Ringemann et al. 2008); 3) focal laser intensities two or more orders of magnitude higher than the ones used in the present study would be needed to produce local heating or trapping effects (Eggeling et al. 2009); 3) the red-edged excitation at ~640 nm wavelength, as used in the present work, is poorly absorbed by the cell (Mueller et al. 2011), thus minimizing phototoxic effects; and most importantly, 4) prolonged exposures in the same area would be required to generate local photoinduced effects (Donnert et al. 2007; Donnert et al. 2009) – yet, it is clear that MINFLUX in single-particle tracking mode as used in the present work, definitely does not meet conditions that would lead to adverse effects, since the patterned doughnut-shaped beam is constantly moved to illuminate around the estimated emitter position, and rapidly refocused and zoomed in onto the moving molecule to “corral” and “lock” it to follow its trajectory, as outlined in the Introduction section of the manuscript.

C. Eggeling, J. Widengren, R. Rigler, C. A. M. Seidel, "Photobleaching of fluorescent dyes under conditions used for single-molecule detection: Evidence of two-step photolysis", *Anal Chem* 1998, 70, 2651-2659.

C. Eggeling, A. Volkmer, C. A. M. Seidel, "Molecular Photobleaching Kinetics of Rhodamine 6G by One- and Two-Photon Induced Confocal Fluorescence Microscopy", *Chemphyschem* 2005, 6, 791-804.

A. Dasgupta, A. Koerfer, B. Kokot, I. Urbancic, C. Eggeling, P. Carravilla, "Effects and avoidance of photoconversion-induced artifacts in confocal and STED microscopy", *Nat Methods* 2024, 21, 1171–1174, 10.1038/s41592-024-02297-4.

C. Eggeling, J. Widengren, L. Brand, J. Schaffer, S. Felekyan, C. A. M. Seidel, "Analysis of photobleaching in single-molecule multicolor excitation and forster resonance energy transfer measurement", *J Phys Chem A* 2006, 110, 2979-2995.

C. Ringemann, A. Schonle, A. Giske, C. von Middendorff, S. W. Hell, C. Eggeling, "Enhancing Fluorescence Brightness: Effect of Reverse Intersystem Crossing Studied by Fluorescence Fluctuation Spectroscopy", *Chemphyschem* 2008, 9, 612-624.

C. Eggeling, C. Ringemann, R. Medda, G. Schwarzmann, K. Sandhoff, S. Polyakova, V. N. Belov, B. Hein, C. von Middendorff, A. Schonle, S. W. Hell, "Direct observation of the nanoscale dynamics of membrane lipids in a living cell", *Nature* 2009, 457, 1159-1162, 10.1038/nature07596.

V. Mueller, C. Ringemann, A. Honigsmann, G. Schwarzmann, R. Medda, M. Leutenegger, S. Polyakova, V. N. Belov, S. W. Hell, C. Eggeling, "STED nanoscopy reveals molecular details of cholesterol- and cytoskeleton-modulated lipid interactions in living cells", *Biophys J* 2011, 101, 1651-1660, 10.1016/j.bpj.2011.09.006.

G. Donnert, C. Eggeling, S. W. Hell, "Major signal increase in fluorescence microscopy through dark-state relaxation", *Nat Methods* 2007, 4, 81-86.

G. Donnert, C. Eggeling, S. W. Hell, "Triplet-Relaxation Microscopy with bunched pulsed excitation", *Photochem Photobiol* 2009, 8, 481-485.

2- To better understand the distribution of the nAChR receptor fused to CF640-BTX and the Cho molecules (fPEG-cho) in the plasma membrane, we suggest that the authors include representative images of cells with the double-stained in supplementary material. As mentioned on page 5, the authors inspected at least 10 cells for each experimental condition.

Reply: We thank the reviewer for raising this point. The concentrations of the dyes, as well as the MINFLUX instrumental parameters were optimized for the fastest acquisition possible to interrogate translational motion of the receptor protein in a time window beyond that afforded by camera-based techniques. Single-molecule tracking conditions in live cells in the sub-millisecond range and with minimal photon fluxes are not optimal for static imaging, i.e., to show the distribution of receptor and cholesterol in the cell. A former paper from the Barrantes lab (Kamerbeek et al., Biophysical Journal 2013) provides representative images of fluorescent labelled receptor and fPEG-cholesterol in the very same clonal cell line using wide-field microscopy. In Kamerbeek et al. receptor was labelled with a monoclonal antibody (mAb 210) followed by far red secondary antibody labelled with Alexa-647 at 4 °C. Under these conditions, antibody labelling results in uniform overall membrane staining with additional diffraction-limited puncta of antibody-crosslinked receptor molecules (these correspond to the nanoclusters of Kellner et al., 2007). fPEG-Chol labelling at a concentration as high as 1 μM yielded a uniform distribution of stain restricted to the cell-surface membrane, too.

Figure 1 of the main text in the revised version of our manuscript shows representative single-molecule tracks representative of the cell surface labelling with fluorescent BTX and fPEG-Chol at sub-stoichiometric concentrations (0.5 nM CF[®]680R-BTX and 1 nM fPEG-Chol) optimized for single-molecule tracking. Nonetheless, following the reviewer's suggestion, we have incorporated a new figure in the Supplementary Material showing nAChR labelled at a high concentration of fluorescent BTX (1 μM) imaged in confocal and STED superresolution modes, i.e. at a concentration three orders of magnitude higher than that used in the MINFLUX experiments. The new figure reveals the high density of receptor molecules at the surface of CHO-K1/A5 cells, only a fraction of which is labelled in the MINFLUX experiments. Likewise, 1 nM fPEG-Chol added as an exogenous probe represents a negligible amount of sterol in the membrane, too faint for conventional imaging (not shown). Endogenous cholesterol is deemed to be at close to saturating concentrations in the plasma membrane of nAChR-expressing cells.

Ref.: Kamerbeek et al. (2013) Antibody-induced acetylcholine receptor clusters inhabit liquid-ordered and liquid-disordered domains. *Biophys. J.* 105: 1601-1611.

3- It is suggested to include in the text, at line 106, the excitation wavelengths for both probes, as well as the emission wavelengths that allow discrimination between them by ratiometric measurement.

Reply: Complied with in the revised version of the ms. In addition, a new figure (Figure 7) is included in the revised version of the ms to clarify this point.

4- Review in the text of the supplementary material, line 12 "A dye which emits only in I1 will have a DCR value of 1, and its DCR will be 0." Considering the formula used to calculate the DCR value, if a molecule emits only in I1, its DCR will be 1 and its I2 would be 0.

Reply: We thank the reviewer for addressing this point. Clarified in the revised version (see Suppl. Figure 2 and related text).

5- In Supplementary Figure 1, in the DCR histograms of experiments with simultaneous staining using CF[®]680R-BTX and fPEG-Chol (right), a difference between the two mean distributions can be observed, more markedly in the distribution of fPEG-Chol alone and when co-labeled with CF[®]680R-BTX. Additionally, a broader distribution can be observed in the histograms, indicating the contribution of CF[®]680R-BTX emission signals to fPEG-Chol emission and vice versa. Could you briefly explain the reason for these seemingly significant differences and whether you considered any correction factor to account for false positives in order to isolate trajectories with fPEG-Chol and BTX680R when both experimental labeling conditions were sampled simultaneously?

Reply: A good point. Yet the seemingly significant differences are in fact NOT statistically significant. We measured the proportion of false positives under our selected threshold (0.55) and found that fPEG-Chol had no false positives, while CF[®]680R-BTX had a false positive rate of 1%. The apparent significant separation between the two probes when sampled in isolation suggests that when they are sampled simultaneously, they are almost always correctly classified. Moreover, since we take the average of the DCR values from a given trajectory to classify them, the classification is even more precise. This becomes more evident in the revised version of Supplementary Figure 2, where we have coloured the relevant areas making it now possible to see that the DCR distributions from either fluorophore in isolation do not overlap, and that the thresholds are valid when the emitters are present simultaneously. We have further clarified this point in the revised Supplementary Material, and analysed the difference between the two distributions, which are significantly different from the statistical standpoint ($p < 0.001$).

6- In Table 1, in the section Perturbation of the submembrane actin cytoskeleton network by CK-666, Far-red toxin (CF[®]680R-BTX)-labelled nAChR upon CK-666 treatment results are not found (Far-red toxin (CF[®]680R-BTX)-labelled nAChR upon CK-666 treatment). Review and ensure that the nomenclature is consistent with the rest of the table.

Reply: We thank the reviewer for raising this apparent incongruency. The table was complete, but the nomenclature did not coincide with that in the main text. We apologise for this. We have now moved the section on CK-666 treatment to the Suppl. Material (Suppl. Table 2), which now shows the far-red toxin (CF[®]680R-BTX)-labelled nAChR upon CK-666 treatment in the second part of the table as "*Perturbation of the submembrane actin cytoskeleton network by CK-666*".

7- In line 202, If you are referring to other studies where SMT analysis has reported β threshold values higher than those in this work for trajectory classification, it missing the references.

Reply: We thank the reviewer for the suggestion. The original sentence was intended to clarify that similar β ranges were used in refs. (Mosqueira et al 2018, 2020; Maizón & Barrantes 2022) to classify superdiffusive behaviour. The point is clarified in the revised version of the ms following the reviewer's suggestion.

Main text (page 10): *The large fraction of superdiffusive trajectories resulted from the rather low threshold set for β (> 1.1) for this population of trajectories, to facilitate comparison with previous SMT analyses (Mosqueira et al. 2018, 2020; Buena-Maizón and Barrantes 2021)). This also enabled us to highlight tendencies towards faster diffusion, e.g., due to larger fractions of more fluid (or less ordered) membrane regions during the track. When analysed according to their diffusive behaviour, subdiffusive and Brownian particles labelled with either toxin displayed statistically indistinguishable values, while CF[®]680R-BTX superdiffusive particles moved faster than those of CF[®]640-BTX. This is clearly observed in the Suppl. Figure 8 introduced in the revised version, where the histograms showing the distribution of the $K\beta$ values of the two fluorescent toxins are displayed: only a minor proportion (~1%) of outliers corresponding to the superdiffusive tracks of CF[®]680R-BTX-labelled receptors are singled out. It is therefore the weight of the superdiffusive component that distorted the apparent average value of the ensemble population. Moreover, regardless of whether CF[®]640R-BTX or CF[®]680R-BTX(+fPEG-Chol) was used, the two probes exhibited similar trends, and accelerated diffusion with increasing concentrations of exogenous cholesterol; the tendencies remained consistent for the other dynamic parameters (Table 1 and Suppl. Table 2).*

8- Previous studies have reported the lateral diffusion of transmembrane receptors, demonstrating heterogeneous diffusion behavior. It has also been proposed that interactions with the cytoskeletal mesh or lipid domains could constrain membrane receptor movement in the plasma membrane (PM) using various fluorescence microscopy techniques. In this work, the study by Mosqueira et al. (2018, 2020) has been cited, reporting that the translational motion heterogeneity of the nAChR is characterized by the coexistence of dispersed individual macromolecules and nanoclusters of variable sizes. Considering that these nanoclusters or nanodomains are highly dynamic structures, a broader discussion based on the results of this study would be relevant.

Reply: The suggestion is well taken, and additional discussion is included in the revised version of the ms.

Main text (page 24): *From our previous camera-based SMT results at lower time resolution we proposed that crowding of nAChRs in nm-sized aggregates (nanoclusters) may impede the motion of the individual receptor macromolecules (Mosqueira et al. 2018, 2020). We further proposed a picket-like mechanism to account for the transient motional hindrance. The experimental data could be accounted for in terms of a 2-state model (Grebekov 2019) in which receptors switched between Brownian motion and obstructed diffusion (OD) (Mosqueira et al. 2018, 2020). The present results clearly show that the restricted diffusion sojourns also occur in the fast (millisecond and sub-millisecond) time domain accessible to MINFLUX. A deep learning approach (Buena-Maizón and Barrantes 2021) concurred with the early proposal. Using concatenated convolutional neural networks, a more recent analysis challenged the OD model against six other physical models; the 2-state model withstood the challenge and remains the simplest interpretation of the translational diffusion of the nAChR at the cell surface (Saavedra and Barrantes 2025), a contention that appears to be extensive to the present experimental work.*

9- Correct the definition of (PDF) in Figure 3. If possible, simplify Figure 3, as it contains a large

amount of information. Consider presenting only the probability density functions corresponding to fewer step lags.

Reply: We thank the reviewer for the suggestion. To simplify the figure, we decided to remove PDFs from step lags 25 and 50 in all experimental conditions in Figure 3 and (new) Suppl. Figure 6 of the revised version.

10- Introduce the full meaning of the acronym MINFLUX the first time it appears in the text.

Reply: done.

Reviewer #2 (Remarks to the Author):

Dear Editor,

The manuscript entitled “Concurrent diffusion of nicotinic acetylcholine receptors and fluorescent cholesterol disclosed by two-colour sub-millisecond MINFLUX-based single-molecule tracking” by Reina et al., presents a study of a neurotransmitter receptor at the plasma membrane of mammalian cells, using a modern and high-end technique (MINFLUX microscopy). The experiments are well performed and explained in detail, the conclusions are properly supported by the results, and the manuscript is in general well written. Considering the fast development of these kind of techniques, providing a higher spatial and temporal resolution, this study should appeal a broad reader community. However, some concerns must be addressed, before the manuscript can be recommended for publication:

The authors have vast experience in this system, and have already presented studies using other established techniques such as SPT-STORM, FCS, FRAP (e.g. Eur Biophys J (2010) 39:213–227, Almarza 2014, Mosqueira 2018). Thus, in my opinion, the main novelty of the manuscript is the successful application of MINFLUX tracking to this system for the first time. As this technique offers a higher spatial and temporal resolution, this provides the possibilities to reveal phenomena otherwise hidden in other classical techniques. The manuscript could be improved by using the Discussion section to describe the advantages observed (not of the technique itself, but the advantages observed in practice) and for a close comparison with results previously obtained with methods of lower spatiotemporal resolution. In particular, if there are some new phenomena that was not observed before, if there are experiments or data analysis performed for the first time here, that were not possible before due to limitations of classical techniques, and which results and measured parameters agree or not with previous reports, or are reported here with a higher accuracy. The reader should be able to assess this information without the need to reach for a larger number of references.

Reply: We thank the reviewer for raising this point, which is addressed following the reviewer’s suggestion in the revised version of the manuscript.

Other concerns:

1) Did previous studies (from this or other authors) performed two-color SPT, FCS (or any other technique) on the same system? If this is the case, the results should be also compared, and if not, the novelty of this work regarding this aspect should be highlighted. As it reads, it is clear this is the first study presenting two-color MINFLUX tracking using spectral discrimination, but it is not clear if this is the first co-tracking experiment of this receptor by any method.

Reply: We thank the reviewer for highlighting this point. As stated throughout the paper, to the best of our knowledge this is the first study of its kind, in the two aspects mentioned by the reviewer. We have now emphasized this in the main text of the revised manuscript and in the last sentence of the Abstract: *“To the best of our knowledge, this study constitutes the first series of experiments showing the diffusion dynamics of a transmembrane protein -a functionally important neurotransmitter receptor- together with a key membrane lipid in the native plasma membrane of a live cell and is also the first to address this topic using two-colour MINFLUX with spectral discrimination.”*

2) The dyes used are referred as CF640/CF680 and later CF640R/CF680R, but these are different dyes. According to the vendor CF680 is a cyanine-based dye and CF680R a rhodamine-based dye, both with quite different properties. CF680 has been used for MINFLUX imaging, in part due to its tendency to blink, which is undesired for tracking. The actual dyes used should be clarified, and eventually any issues with blinking discussed.

Reply: We only used the rhodamine-based CF probes, not the cyanine-based probes. The missing letter R is a typo that has been amended in the revised version. Apologies. We did not perform MINFLUX-based imaging, but only tracking, where no blinking is needed.

3) Intuitively, the probes are added at a very low concentration and should not affect or modify the system, or at least have a minimal perturbation. However, results differ when a different probe is used: for instance, CF640R-BTX and CF680R-BTX results in table 1, or “the motion of the nAChR, as assessed by its apparent anomalous diffusion constant K_{β} , was influenced by the presence of the fluorescent cholesterol probe” (page 25). Was there any attempt to lower the concentration of the probes?

Reply: The reviewer is right in pointing out that a “probe” should not affect or modify the system. But one must bear in mind that the fluorescent toxins are not in fact canonical probes; they are high-affinity, quasi-irreversible competitive antagonists of nicotinic receptors, i.e., receptor ligands with distinct pharmacological activity! Hence, we considered the possibility that the chemical structures of the two fluorophores (CF640R and CF680R) differ. We therefore asked the manufacturers (Biotium). They replied that the (chemical) “structure of the fluorophores is proprietary”, and hence we ignore whether the CF-labelled toxin adducts could differ in such a way as to exhibit different affinities for the receptor. The only information available is that the two fluorophores in aqueous media have different fluorescence lifetimes, but the lifetimes upon conjugation to BTX are not available from the manufacturers. This photophysical property is, however, unlikely to affect the translational motion of the toxin-tagged receptor macromolecule.

The relevant point is that the two rhodamine derivative-labelled toxins respond in an equivalent fashion in multiple parameters to cholesterol depletion/increases. That is, they sense the receptor motional properties in a similar manner. However, attending to the concern raised by the reviewer, we have added a new figure (Suppl. Figure 8) in the revised submission, showing the distribution of the K_{β} values of CF[®]640R-BTX and CF[®]680R-BTX. The histogram clearly shows that in fact the general diffusion constant values of the two fluorescent toxins exhibit a large degree of overlap, except for a minor component (ca. 1%) stemming from CF[®]680R-BTX-labelled nAChRs that diffuse at much higher rates. This can also be appreciated in Table 2 where the higher percentage of superdiffusive CF[®]680R-BTX molecules is apparent. The difference between the two distributions has now been challenged analytically and found to be statistically very significant ($[<0.001]$).

Regarding the second point -concentration of the probes- the dye labelling and other experimental conditions were optimized for the fastest acquisition possible to interrogate translational motion of the receptor protein in the sub-millisecond range and with minimal photon fluxes, i.e., adapted to MINFLUX implicit requirements. After several trials, fluorescent BTX and fPEG-Chol concentrations were set at sub-stoichiometric levels (0.5 nM CF[®]680R-BTX and 1 nM fPEG-Chol), about *three orders of magnitude below the concentrations used for static imaging* of molecule topography (now shown in Suppl. Figure 2 in response to the other reviewer's suggestion). Only a small percentage of the receptors were labelled under such conditions, optimized for tracking. The fluorescent cholesterol analogue was used at a concentration of the same order of magnitude as that of the fluorescent BTX, which corresponds to an extremely low fluorescent cholesterol: unlabelled endogenous cholesterol ratio. This also discards crosstalk effects between fluorescently labelled molecules in the crowded membrane environment.

4) Regarding the quantitative analysis of overlap between labelled nAChR and Chol trajectories: do the results depend on the concentration of the fluorescent probes used for labelling? Considering the low concentration of the probes, a 70 % trace overlap seems too high. Is there an estimation of the percentage of nAChR receptors and Cholesterol molecules labelled? Are all receptors labelled.

Reply: We set the concentration of the probes very low to avoid any bias to MINFLUX-based single-molecule tracking (as is done in most single-molecule tracking applications). Therefore, we have not done a concentration series. Yet, it would be interesting to see the effect on the (especially two-colour) tracking results, upon moving to higher and potentially biasing concentrations. However, this is a more technical issue and will be the topic of a future series of experiments.

Regarding the second issue raised in point (4), we never stated that the trace overlap is 70%. In line 345, we wrote "70% of the ROIs were found to have statistically significant overlap". The confined and non-confined traces overlap, given by the overlap coefficient C , is ~ 0.27 (in other words, 27% of the confined traces between nAChR and Chol overlapped) under control conditions. Increasing the exogenous cholesterol concentration resulted in a lower degree of overlaps in both confined and non-confined portions of the trajectories. All reported values in the paper are the average of averages of ROIs. We followed the recommendations outlined in SuperPlots (Lord et al., 2020), aimed at increasing statistical robustness. We apologise for not having explicitly stated this in the statistical analysis section. Amended in the revised version, with inclusion of the relevant reference.

5) In the colour discrimination (DCR analysis), a threshold of 0.55 is defined in Supplementary Figure 1 and in figure 4. However, in the main text (pages 15-16), the analysis is based on a double threshold of 0.4 and 0.55, with this central window used for colocalization. This should be clarified.

Reply: When both nAChR and Chol trajectories coexisted in the experiments, they were classified according to their mean DCR (as detailed in the first section of the Supplementary Material and its Suppl. Figure 1). In the main text pages 15-16, we did not define a double threshold, but a range from 0.4 to 0.55, a central *window* manually selected where colocalization of the two probes may occur as the DCR significantly drops (in the case of fPEG-Chol) or increases (in the case of toxin-labelled nAChR). We have now adapted Supplementary Figure 2, adding a second threshold line at 0.4. Further, we have added three shaded portions

on the right-side graph, one for localization assigned to BTX CF680R, one for fPEG-Chol, and one for colocalization.

Reviewer #3 (Remarks to the Author):

Reply: We thank the reviewer for taking time to critically and constructively assess our manuscript and work.